# Clp protease and antisense RNA jointly regulate the global regulator CarD to mediate mycobacterial starvation response

Xinfeng Li[1†], Fang Chen[1†], Xiaoyu Liu[1], Jinfeng Xiao[1], Binda T Andongma[1], Qing Tang[1], Xiaojian Cao[1], Shan-Ho Chou[1], Michael Y Galperin[2], Jin He[1]*

[1]State Key Laboratory of Agricultural Microbiology & Hubei Hongshan Laboratory, College of Life Science and Technology, Huazhong Agricultural University, Wuhan, China; [2]National Center for Biotechnology Information, National Library of Medicine, National Institutes of Health, Bethesda, United States

*For correspondence:
hejin@mail.hzau.edu.cn

[†]These authors contributed equally to this work

Competing interest: The authors declare that no competing interests exist.

**Abstract** Under starvation conditions, bacteria tend to slow down their translation rate by reducing rRNA synthesis, but the way they accomplish that may vary in different bacteria. In *Mycobacterium* species, transcription of rRNA is activated by the RNA polymerase (RNAP) accessory transcription factor CarD, which interacts directly with RNAP to stabilize the RNAP-promoter open complex formed on rRNA genes. The functions of CarD have been extensively studied, but the mechanisms that control its expression remain obscure. Here, we report that the level of CarD was tightly regulated when mycobacterial cells switched from nutrient-rich to nutrient-deprived conditions. At the translational level, an antisense RNA of *carD* (AscarD) was induced in a SigF-dependent manner to bind with *carD* mRNA and inhibit CarD translation, while at the post-translational level, the residual intracellular CarD was quickly degraded by the Clp protease. AscarD thus worked synergistically with Clp protease to decrease the CarD level to help mycobacterial cells cope with the nutritional stress. Altogether, our work elucidates the regulation mode of CarD and delineates a new mechanism for the mycobacterial starvation response, which is important for the adaptation and persistence of mycobacterial pathogens in the host environment.

## Editor's evaluation

CarD is an RNA polymerase interacting protein that is essential for mycobacterial viability, the levels of which are important for controlling gene expression in mycobacteria during various stress conditions. This study reports two mechanisms that regulate levels of CarD under stress conditions, including starvation. The authors report that CarD levels are tightly regulated and that there was a dramatic decrease in the levels of CarD when cells switched from the nutrient-rich to the starvation condition. They discovered two synergistic mechanisms that led to this dramatic decrease in CarD. The first is SigF-dependent induction of antisense RNA of CarD (AscarD), which inhibits CarD translation and a second mechanism involving Clp protease-mediated degradation of intracellular CarD. The work will be of interest to researchers studying non-coding RNAs, microbial gene expression, physiology and stress response.

## Introduction

Bacterial starvation response refers to the physiological changes occurring in bacteria due to the lack of external nutrients during their growth and reproduction (*Morita, 1982*). Under starvation conditions, bacterial cells usually reduce the synthesis of rRNA and ribosome proteins (*Gourse et al., 1996*; *Paul et al., 2004*). The mechanisms of starvation response that have been elucidated in such bacteria as *Escherichia coli* and *Bacillus subtilis* work primarily by reducing rRNA transcription via decreasing the stability of the transcription initiation complex (*Gourse et al., 2018*; *Hauryliuk et al., 2015*).

*Mycobacterium* is a widespread genus of Gram-positive bacteria that comprises several important pathogens, including *Mycobacterium tuberculosis*, the causative agent of tuberculosis, which kills ~1.5 million people every year. One of the main difficulties in eliminating *M. tuberculosis* is that it usually responds to various host stresses, such as nutritional starvation, low oxygen, and low pH, by entering into a dormant state, which renders the organism extremely resistant to host defenses (*Gengenbacher and Kaufmann, 2012*). This genus also includes nonpathogens, such as *M. smegmatis*, which is widely used as a model organism for mycobacterial research. At present, the starvation response mechanisms of mycobacterial cells remain obscure.

Mycobacterial RNA polymerase (RNAP) is usually less efficient in forming RNAP-promoter open complex (RPo) than *E. coli* RNAP on the rRNA genes (*Davis et al., 2015*), and the RPo formed is rather unstable and readily reversible (*Davis et al., 2015*; *Rammohan et al., 2015*). To overcome this deficiency, mycobacterial cells have evolved two accessory transcription factors, CarD and RbpA, that help RNAP form a stable RPo (*Hubin et al., 2017*; *Jensen et al., 2019*; *Rammohan et al., 2016*). Both are global transcription factors that interact directly with RNAP to regulate the transcription of many downstream genes, including those of rRNA (*Rammohan et al., 2016*; *Sudalaiyadum Perumal et al., 2018*; *Zhu et al., 2019*). CarD stabilizes mycobacterial RPo via a two-tiered kinetic mechanism. First, CarD binds to the RNAP-promoter closed complex (RPc) to increase the rate of DNA opening; then, CarD associates with RPo with a high affinity to prevent the DNA bubble collapse (*Davis et al., 2015*; *Hubin et al., 2017*; *Rammohan et al., 2015*). Although binding of CarD to RNAP tends to increase the stability of RPo, it may also delay the dissociation of RNAP from the promoter region and thus hinder transcription progress (*Jensen et al., 2019*). Therefore, CarD may also inhibit the expression of certain genes. Whether CarD activates or inhibits the expression of a specific target gene appears to be determined by the kinetics of the initiation complex formation among CarD, RNAP, and the specific promoter (*Jensen et al., 2019*; *Zhu et al., 2019*). CarD was found to be essential for the survival of mycobacterial cells (*Stallings et al., 2009*) and weakening the interaction between CarD and RNAP rendered mycobacterial cells more sensitive to oxidative stress, DNA damage, and the effect of some antibiotics (*Garner et al., 2014*; *Stallings et al., 2009*; *Weiss et al., 2012*). A recent study showed that CarD regulates (either activates or inhibits) the expression of approximately two-thirds of genes in *M. tuberculosis* (*Zhu et al., 2019*). Despite the fact that CarD plays such a critical role in mycobacteria, the mechanisms that regulate its cellular levels remain largely uncharacterized.

It is worth noting that CarD was initially thought to inhibit the transcription of rRNA genes, and the transcription of *carD* was upregulated in response to starvation (*Stallings et al., 2009*). However, more recently, it was reported that CarD is a transcriptional activator of rRNA genes (*Rammohan et al., 2015*; *Srivastava et al., 2013*) and the growth rates of mycobacterial cells positively correlate with the CarD content (*Garner et al., 2017*; *Stallings et al., 2009*; *Weiss et al., 2012*). Nevertheless, the expression of CarD is still considered to be upregulated in response to starvation. If this was the case, the increased CarD would accelerate rRNA synthesis and mycobacterial growth under the starvation condition, which seems to contradict the current consensus (*Irving and Corrigan, 2018*; *Rasouly et al., 2017*; *Srivatsan and Wang, 2008*). Therefore, it is important to clarify the regulation of CarD expression under starvation conditions. In the current study, we found that although *carD* transcript levels were upregulated in response to starvation, its protein levels dramatically decreased. Further, we found that the reduction of CarD protein level under starvation conditions is a common regulatory mechanism that depends upon the functioning of both antisense RNA and Clp protease. This study describes the mechanisms behind the apparent contradiction between CarD mRNA and protein levels and reveals a new mechanism of mycobacterial response to stress.

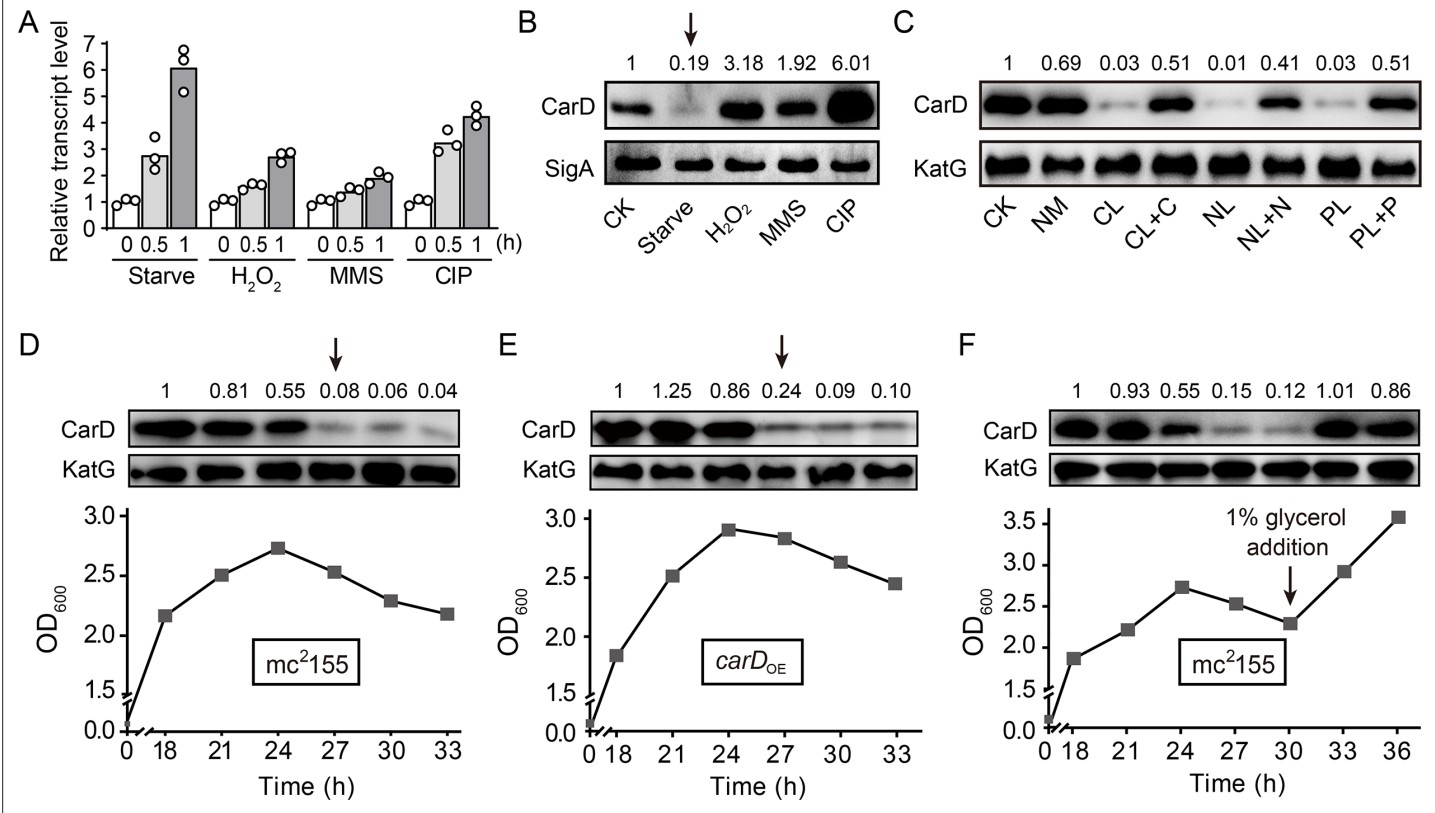

**Figure 1.** Changes of CarD transcript and protein levels under starvation and genotoxic stress. (**A, B**) The transcript and protein levels of CarD, respectively, under different stress conditions. The *carD* transcript levels in the treated exponential mc²155 cells were measured by qRT-PCR, normalized to the *sigA* transcript levels, and expressed as the fold change of untreated cells. CK indicates the untreated cells of mc²155. 'Starve' means that mc²155 cells were first cultured in the 7H9 medium, and then transferred to phosphate-buffered saline (PBS) for 0.5 or 1 hr. For stimulation experiments, 10 mM $H_2O_2$, 0.1% methyl methanesulfonate (MMS), and 10 µg/ml of ciprofloxacin (CIP) were used. Individual data for the three biological replicates are shown in the corresponding columns. Western blot was used to detect the CarD protein levels under the same treatment conditions with SigA serving as the internal reference protein. (**C**) Protein levels of CarD under distinct starvation conditions. CK indicates the untreated exponential cells; NM indicates the exponential cells transferred into the normal medium for 4 hr; CL, NL, and PL indicate the exponential cells transferred into carbon-, nitrogen-, and phosphorus-limited media for 4 hr, respectively; CL + C, NL + N, and PL + P indicate the starved mc²155 cultures supplemented with the corresponding nutrients for 4 hr. KatG was used as the control in the Western blot experiments. CarD protein levels at the different growth stages in mc²155 (**D, F**), and *carD* overexpressing strain (*carD*OE, panel **E**). The lower part of the chart shows the respective growth curves with the sampling times. For panels (**B–F**), the number above each band of the Western blot represents their relative quantitative values, which are normalized with respect to their corresponding loading controls. For panels (**B, D, and E**), arrows above the Western blot results indicate the sharp decrease in CarD levels under starvation or stationary phase.

The online version of this article includes the following source data and figure supplement(s) for figure 1:

**Source data 1.** Changes of *carD* transcript levels under starvation and genotoxic stress (numerical data for *Figure 1A*).

**Figure supplement 1.** Changes of *carD* levels in *M. smegmatis* under different conditions and different strains.

## Results

### CarD protein level increases under genotoxic stresses but dramatically decreases under starvation conditions

CarD is an essential RNAP-interacting protein that regulates the transcription of rRNA genes and many related genes by stabilizing the RPo. While Stallings et al. found that the *carD* gene is upregulated in response to starvation and genotoxic stresses in *M. smegmatis* strain mc²155 (*Stallings et al., 2009*), they only monitored the transcriptional level but not the translation of *carD*, which may not truly reflect the CarD protein content. Therefore, to clarify the dynamics of CarD content under the starvation condition and genotoxic stresses, we examined both the *carD* transcript and CarD protein levels in the mc²155 strain by quantitative real-time PCR (qRT-PCR) and Western blot experiments,

respectively. As shown in *Figure 1*, panels A and B, both *carD* transcript and CarD protein levels increased under genotoxic stresses triggered by $H_2O_2$, methyl methanesulfonate (MMS), or ciprofloxacin (CIP), which was consistent with the previous reports that CarD may be involved in DNA damage repair (*Stallings et al., 2009*). However, although the *carD* transcript level increased in response to starvation (*Figure 1A*), the CarD protein level, instead, decreased (marked by an arrow in *Figure 1B*). This observation indicates that the *carD* transcript level is upregulated in response to starvation, as previously reported (*Stallings et al., 2009*), but there are other mechanisms that down-regulate CarD protein level.

To investigate whether the decline in the CarD level is due to the lack of a specific nutrient or to a general response to starvation stress, we investigated the changes in CarD levels under carbon-, nitrogen-, and phosphorus-starvation conditions. We first cultured mc²155 cells to mid-exponential phase (MEP), harvested the cells, and then transferred these cells to the normal medium, carbon-limited, nitrogen-limited, and phosphorus-limited medium, followed by detecting the respective mRNA and protein levels of CarD. It is worth noting that although the *carD* transcript level increased in response to starvation conditions (*Figure 1—figure supplement 1A*), the CarD protein level decreased (*Figure 1C*). When the nutrient-limited media were supplemented with the corresponding nutrients, CarD returned to normal levels.

Since the mycobacterial cells in the stationary phase are in the state of nutritional starvation (*Smeulders et al., 1999*), we also measured the CarD protein levels at different growth periods of mc²155 cells. As shown in *Figure 1D*, the CarD level remained relatively constant in the exponential phase but dropped sharply in the early stationary phase (marked by an arrow in *Figure 1D*), which is consistent with the above starvation experiments. To further verify this result, we constructed a *carD* overexpressing strain (*carD*OE) (*Figure 1—figure supplement 1B–D*) and measured CarD protein levels at different growth periods. Interestingly, despite *carD* overexpression, the CarD protein level still decreased dramatically when the mycobacterial cells entered the stationary phase (marked by an arrow in *Figure 1E*). Since the carbon source in the culture medium was likely depleted when the mycobacterial cells entered the stationary phase (*Smeulders et al., 1999*), we speculated that the decrease in the CarD protein level could be caused by carbon starvation. To verify this hypothesis, we added 1% glycerol (glycerol is the main carbon source under normal culture conditions of *M. smegmatis*) to the mc²155 culture at the stationary phase and measured the CarD protein level 3 and 6 hr thereafter. As shown in *Figure 1F*, the CarD level significantly increased after the glycerol addition, and the mc²155 cells resumed normal growth. Considering that CarD activates the transcription of rRNA (*Rammohan et al., 2015*; *Srivastava et al., 2013*), and that cells need to reduce rRNA levels in response to starvation (*Gourse et al., 2018*), we believe that the reduction in the CarD level under starvation conditions may be an adaptive response of mycobacterial cells. Yet, when nutrients became available, CarD quickly returned to its normal level to allow the cells to resume growth.

## CarD levels are dramatically decreased in *M. bovis* BCG and *M. tuberculosis* under host-like stress conditions

To investigate whether the significant reduction of CarD levels under starvation conditions also happens in other mycobacterial species, we carried out starvation experiments in two other mycobacteria, *M. bovis* BCG and *M. tuberculosis* H37Ra. The results are consistent with those in *M. smegmatis*, that is, CarD levels were all significantly reduced in response to carbon-, nitrogen-, and phosphorus-starvation conditions (*Figure 2A, B*). When nutrient-limited cultures were supplemented with the corresponding nutrients, CarD returned to the normal levels. In addition, we also measured the CarD levels at different growth phases of the two strains. As shown in *Figure 2C, D*, CarD levels were dramatically decreased when BCG and H37Ra cells entered the stationary phase, which is also consistent with the results in *M. smegmatis*. The above results indicate that the rapid reduction of the CarD level in response to starvation is a common phenomenon in mycobacteria, and regulating CarD content to cope with nutritional starvation is a conserved mechanism for the mycobacterial adaptive response.

It is important to note that after infecting the host, pathogenic mycobacterial cells not only suffer from nutritional deprivation but are also exposed to hypoxic and acidic conditions. Therefore, to explore whether CarD plays a role in the adaptation of mycobacterial cells to the host environment, we measured CarD levels under these conditions. As shown in *Figure 2E*, CarD levels significantly

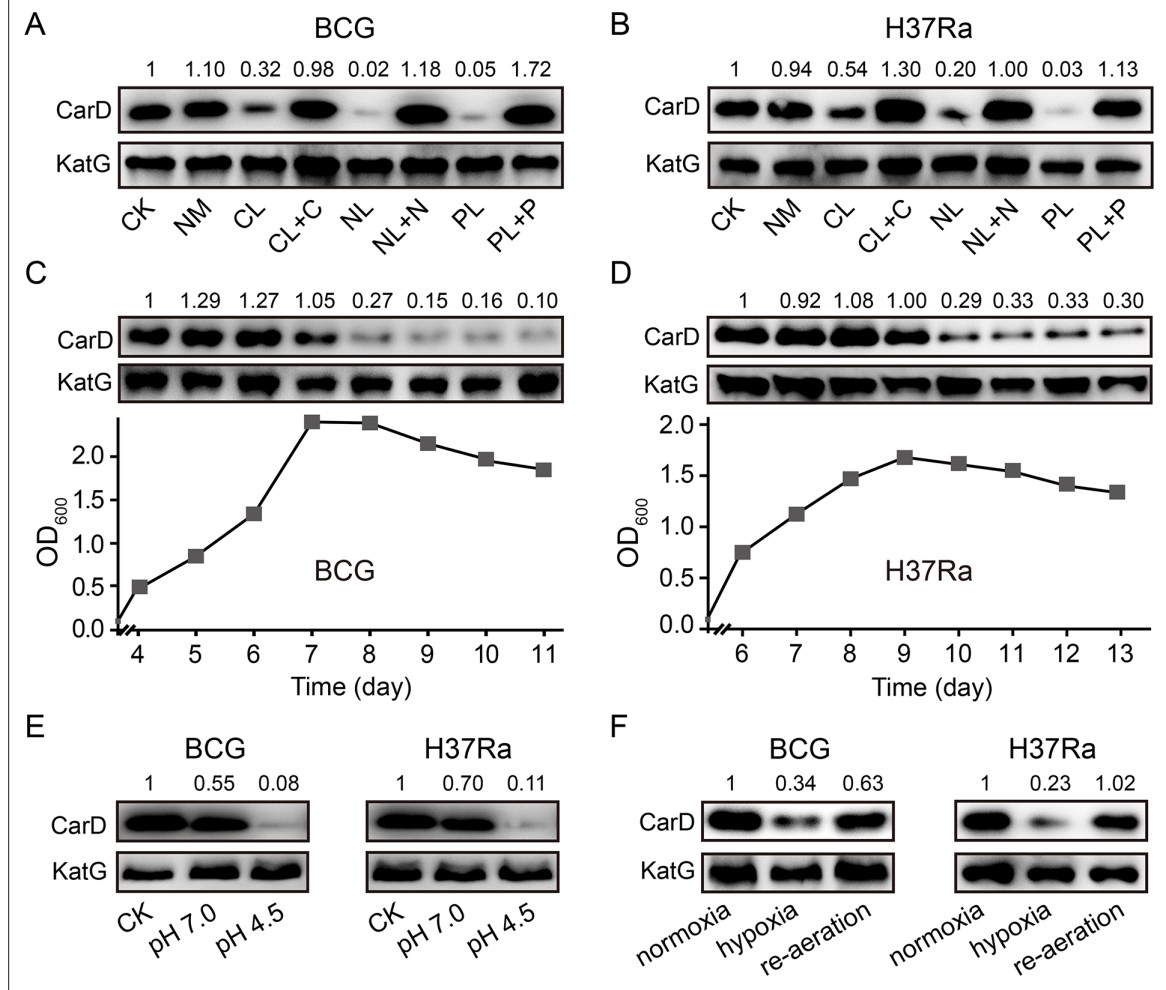

**Figure 2.** Changes of CarD levels in *M.bovis* BCG and *M. tuberculosis* H37Ra under host-like stress conditions. (**A, B**) The protein levels of CarD in BCG and H37Ra strains, respectively, under distinct starvation conditions. (**C, D**) CarD protein levels at the different growth stages of BCG and H37Ra, respectively. (**E**) CarD protein levels in BCG and H37Ra under different pH conditions. (**F**) CarD protein levels in BCG and H37Ra under different oxygen availability conditions. For all panels, the number above each band of the Western blot represents their relative quantitative values, which are normalized with respect to their corresponding loading controls.

The online version of this article includes the following figure supplement(s) for figure 2:

**Figure supplement 1.** Changes of CarD levels in *M. smegmatis* under host-like stress conditions.

decreased when the mycobacterial cells were transferred to the low pH media. For the hypoxic conditions, similarly, CarD levels were also reduced when mycobacterial cells were under hypoxic stress and returned to normal after the cultures were reaerated (*Figure 2F* and *Figure 2—figure supplement 1*). These results suggest that mycobacterial cells reduce CarD levels in response to host stresses to slow down their translation and metabolic rates, which likely contributes to the adaptation of pathogenic mycobacteria to the hostile environment.

## Clp protease degrades CarD under starvation conditions

Since the CarD protein level decreased dramatically under nutritional starvation, hypoxic, and acidic conditions, we speculated that CarD might be proteolytically degraded. Clp is a special energy-dependent protease that regulates the response to various stresses (*Michel et al., 2006*; *Raju et al., 2012*; *Schultz et al., 2017*). The typical Clp proteolytic complex is formed by the association of ClpP, the main proteolytic unit, with an AAA+ (ATPases associated with a variety of cellular activities) unfoldase, either ClpX or ClpA/ClpC (*Kirstein et al., 2009*). Unlike most other bacteria, mycobacteria harbor two ClpP isoforms (ClpP1 and ClpP2), which associate with each other to form the ClpP1P2

heterotetradecamers (*Akopian et al., 2012*; *Li et al., 2016*). Through a quantitative proteomics approach, Raju et al. found that the CarD protein level in the *clpP2* conditional deletion mutant was upregulated (*Raju et al., 2014*). However, that study only measured CarD in the exponential phase, not in the stationary phase. Therefore, it was unclear whether Clp protease mediates the efficient degradation of CarD in the stationary phase. To address this question, we constructed a *clpP2* conditional mutant (*clpP2*CM) (*Figure 3—figure supplement 1*) through the CRISPR/Cpf1-mediated gene editing strategy (*Yan et al., 2017*), in which *clpP2* could be expressed normally only upon addition of 50 ng/ml anhydrotetracycline (ATc), but could not do so when ATc was absent.

To explore the role of Clp protease in CarD degradation, we conducted ClpP2 depletion experiments. The cells of the *clpP2*CM mutant and control cells (Ms/pRH2502-*clpP2*) were first cultured in ATc-containing medium to the late exponential phase ($OD_{600} \approx 1.5$), then harvested, washed, and reinoculated in the fresh medium with or without ATc. The results showed that CarD was effectively degraded when the control cells entered the stationary phase, regardless of the presence of ATc (*Figure 3A, B*). In the *clpP2*CM strain, CarD was also effectively degraded in the stationary phase when ATc was added to induce the *clpP2* expression (*Figure 3C*) but persisted when *clpP2* was not induced (*Figure 3D*). These results indicate that ClpP2 was essential for the efficient degradation of CarD in the stationary phase.

To investigate whether Clp protease is also required for degrading intracellular CarD under starvation conditions, we carried out a series of starvation experiments on *clpP2*CM cells harvested from the MEP. The results showed that CarD was effectively degraded when the ATc-induced *clpP2*CM cells were starved in PBS for 4 hr, while in the ATc-uninduced *clpP2*CM cells CarD was not degraded (*Figure 3—figure supplement 2A, B*). This result is consistent with the experimental data described above, allowing us to conclude that Clp protease was responsible for the degradation of CarD under starvation conditions.

Moreover, mycobacterial CarD contains a highly conserved C-terminal 'LAAAS' sequence (*Figure 3E*), which is similar to the Clp protease recognition motif (*Gallego-García et al., 2017*; *Hoskins and Wickner, 2006*; *Lunge et al., 2020*). To study whether this region mediates the degradation of CarD by Clp protease under stress conditions, we deleted the 'AAAS' coding sequence from the *M. smegmatis carD* gene and checked the CarD protein levels under stationary phase and starvation conditions. The results show that CarD in mc²155 is almost completely degraded under stress conditions, while CarD in the 'AAAS' deletion mutant (AAAS_del) is still highly retained (*Figure 3F, G*). This indicates that the deletion of the 'AAAS' motif largely prevented the Clp protease from degrading CarD. These results further strengthen the notion that Clp protease degrades CarD under starvation conditions.

Additionally, the efficient degradation of large proteins by Clp protease requires their unfolding in the presence of an $AAA^+$ unfoldase (*Akopian et al., 2012*; *Schmitz and Sauer, 2014*). Mycobacteria harbor two functional Clp-associated unfoldases, ClpX and ClpC1 (*Li et al., 2016*; *Schmitz and Sauer, 2014*). Previous proteomics data showed that the CarD protein level is significantly upregulated when ClpC1 is depleted, suggesting that CarD is a substrate of ClpC1 (*Lunge et al., 2020*). To further confirm this result, we carried out an additional pull-down assay. The results showed that CarD does interact directly with ClpC1, but not with ClpX (*Figure 3H*). Therefore, we believe that ClpC1 specifically mediated the degradation of CarD. Furthermore, to clarify why CarD was more effectively degraded in the stationary phase, we monitored the protein levels of ClpP1, ClpP2, and ClpC1. The results showed that the protein levels of ClpP1 and ClpP2 were relatively constant throughout the growth phase (*Figure 3—figure supplement 2C, D*), while the level of ClpC1 protein was significantly upregulated during the stationary phase (*Figure 3I*). Since ClpC1 is the ATPase required for CarD recognition, unfolding, and translocation, its content likely determines the degradation efficiency of CarD. Taken together, these results suggest that the increase of ClpC1 level contributes to the efficient degradation of CarD during the stationary phase.

## Starvation induces the transcription of antisense RNA of *carD*

Next, we wanted to know whether the intracellular CarD content is subject to other types of regulation other than degradation by Clp protease. After mining our previously published RNA-seq data of strain mc²155 (*Li et al., 2017*), we identified an antisense RNA transcribed from the antisense strand of the *carD-ispD* operon. As shown in *Figure 4A*, this antisense RNA (named AscarD) is partially

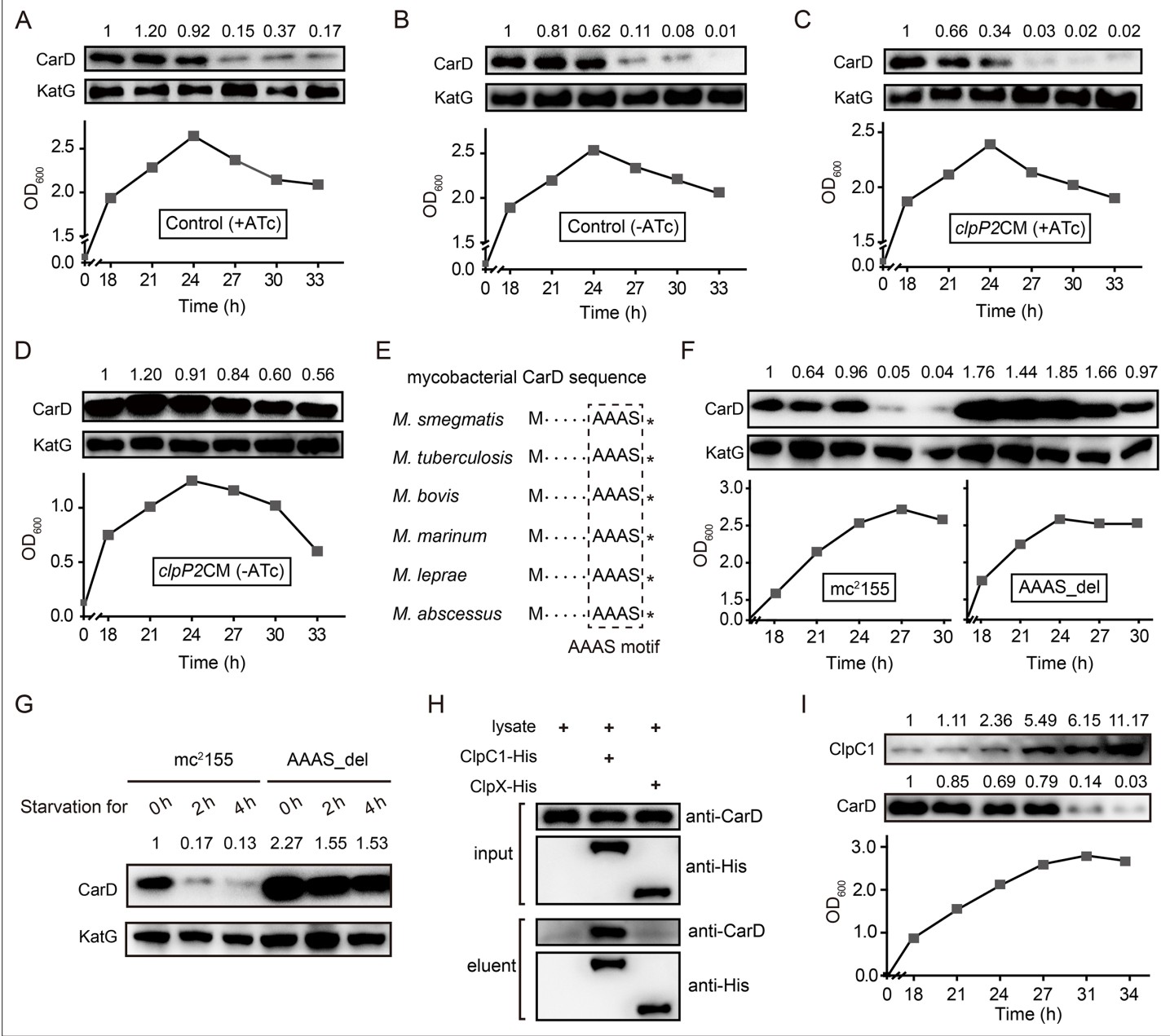

**Figure 3.** Clp protease is responsible for CarD degradation in the stationary phase. (**A–D**) The cells were first cultured in anhydrotetracycline (ATc)-containing medium to the exponential phase (OD$_{600}$ ≈ 1.5), then harvested, washed, and reinoculated in a fresh medium with or without ATc. 0 hr is the time when exponential cells were reinoculated into the fresh medium. (**A, B**) The intracellular CarD levels at different time points of the ATc-induced and ATc-uninduced control cells (Ms/pRH2502-*clpP2*), respectively. (**C, D**) The CarD levels at different time points of the ATc-induced and ATc-uninduced *clpP2*CM (*clpP2* conditional mutant) cells, respectively. KatG was used as the control in the Western blot experiments. (**E**) Conservation of the LAAAS motif in mycobacterial CarD. The asterisk after the LAAAS motif indicates the stop codon. (**F**) CarD protein levels at the different growth stages of mc$^2$155 and AAAS_del cells. (**G**) The starvation experiments of mc$^2$155 and AAAS_del cells. (**H**) Verification of the interaction between CarD and ClpC1/ClpX by pull-down assay. (**I**) Protein levels of ClpC1 and CarD at different growth phases. For panels A–D, F–G, and I, the number above each band of the Western blot represents their relative quantitative values.

The online version of this article includes the following figure supplement(s) for figure 3:

**Figure supplement 1.** Schematic diagram for the construction of the *clpP2* conditional mutant.

**Figure supplement 2.** Clp protease degrades CarD under the starvation condition.

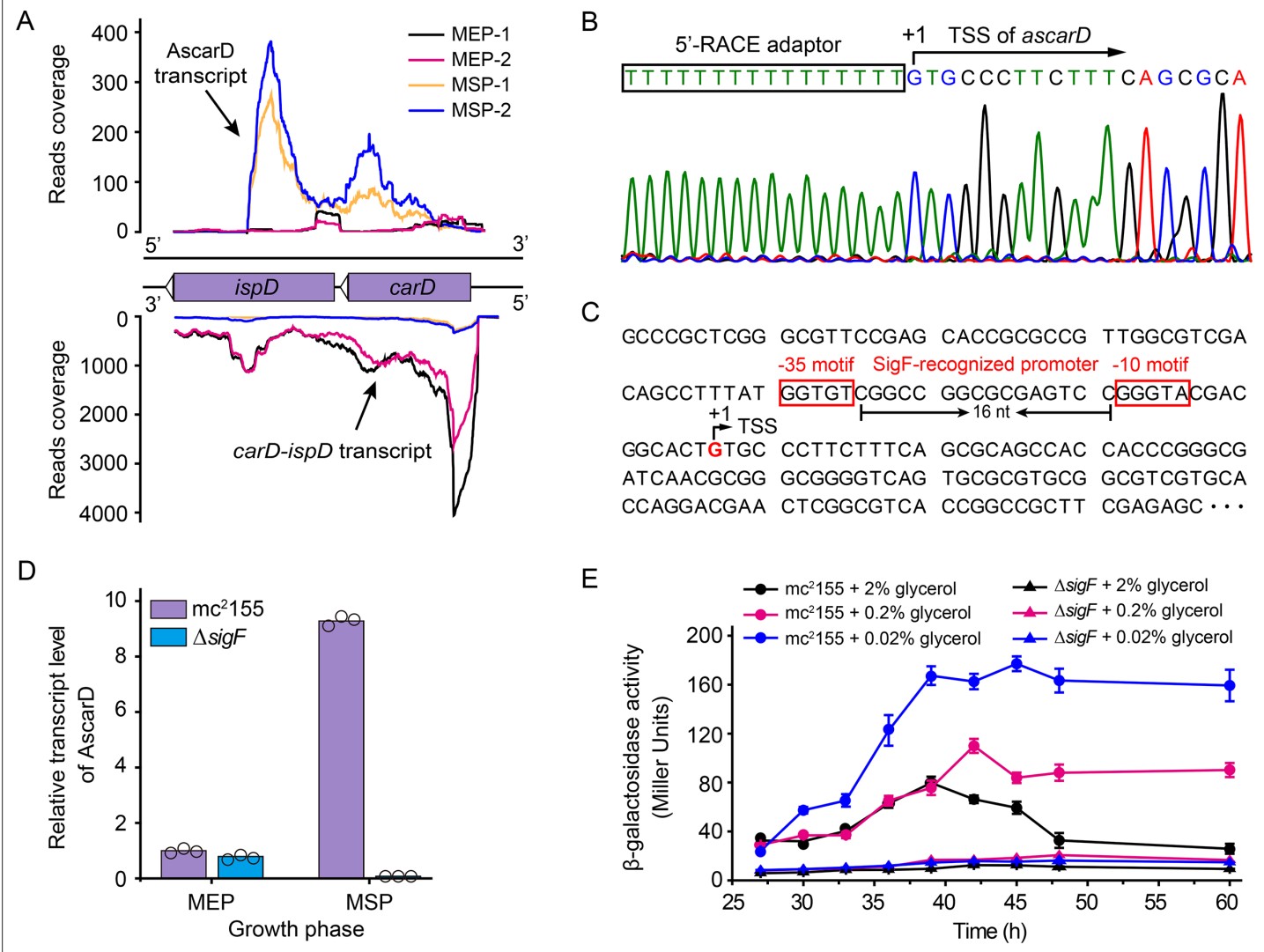

**Figure 4.** Identification and characterization of AscarD. (**A**) Transcriptional landscapes of *carD-ispD* transcript and AscarD. Red and black lines represent exponential-phase cells, blue and green lines are from stationary-phase cells. Extensions of −1 and −2 represent two biological replicates. (**B**) Mapping of the transcriptional start site (TSS) of AscarD. The lower four-color chromatogram shows the results of Sanger sequencing, and the corresponding DNA sequence is displayed on the upper layer. The 5'-rapid amplification of cDNA ends (5'-RACE) adaptor sequence is framed by a black rectangle, and TSS is indicated by a black arrow. (**C**) Potential SigF-recognized −10 and −35 motifs upstream of the identified TSS are indicated with red rectangles. (**D**) AscarD transcript levels at different growth phases of mc²155 and Δ*sigF* strains were measured by qRT-PCR, normalized to *sigA* transcript levels, and expressed as fold change compared to levels of mc²155 cells at mid-exponential phase (MEP). Individual data for the three biological replicates are shown in the corresponding columns. (**E**) Promoter activities of *ascarD* in mc²155 and Δ*sigF* strains carrying a β-galactosidase-encoding reporter plasmid. Error bars indicate the standard deviation of three biological replicates.

The online version of this article includes the following source data and figure supplement(s) for figure 4:

**Source data 1.** AscarD transcript levels at different growth phases of mc²155 and Δ*sigF* strains (numerical data for **Figure 4D**).

**Source data 2.** Promoter activities of ***ascarD*** in mc²155 and Δ*sigF* strains (numerical data for **Figure 4E**).

**Figure supplement 1.** RT-PCR analysis of the transcriptional levels of *ascarD* and *carD*.

complementary to the coding region of *ispD* but fully complementary to the coding region of *carD*. The RNA-seq data also showed that *ascarD* was specifically induced in the mid-stationary phase (MSP) (**Figure 4A**), and we confirmed this by RT-PCR (**Figure 4—figure supplement 1A,B**). Moreover, to determine the specific period when *ascarD* was induced, we examined the RNA level of AscarD throughout the growth phase, and the results showed that *ascarD* was induced at the onset of the stationary phase (**Figure 4—figure supplement 1C**).

To better characterize AscarD, we determined its transcriptional start site (TSS) by carrying out the 5′-RACE (5′-rapid amplification of cDNA ends) experiment (*Figure 4B*). The TSS identified by 5′-RACE was consistent with that revealed by the RNA-seq data. We also discovered potential SigF-recognized −10 and −35 motifs upstream of the identified TSS (*Hartkoorn et al., 2012*; *Hümpel et al., 2010*; *Figure 4C*). SigF is an alternative sigma factor that is active in the stationary phase, which is consistent with the transcriptional pattern of AscarD, suggesting that the transcription of *ascarD* is controlled by SigF. To verify this hypothesis, we examined the transcriptional level of *ascarD* in a *sigF* mutant (Δ*sigF*). As shown in *Figure 4D*, only a very low AscarD level could be detected in the Δ*sigF* strain in the MEP, and transition to the MSP could not induce it either. These data indicate that the expression of *ascarD* is regulated by SigF.

Further, since *ascarD* was highly expressed during the stationary phase, we speculated that transcription of *ascarD* could be also subject to carbon starvation. To verify this idea, we carried out *lacZ* reporter assays to examine the *ascarD* promoter activity under different carbon source (glycerol) concentrations. As shown in *Figure 4E*, the *ascarD* promoter activity gradually increased as the glycerol concentrations decreased. This indicates that *ascarD* could indeed be induced under carbon starvation conditions; however, in the Δ*sigF* strain, the expression of *ascarD* did not respond to the glycerol concentration (*Figure 4E*). This indicates that the response of *ascarD* to low carbon requires the presence of SigF, which is consistent with the above results. Thus, we confirm that the transcription of *ascarD* was highly induced in response to starvation in a SigF-dependent manner.

## AscarD inhibits biosynthesis of CarD protein

Expression of AscarD was highly induced in response to starvation, while the CarD protein level was sharply reduced, suggesting that AscarD could be involved in regulating *carD* expression. To clarify this issue, we carried out *lacZ* reporter assays. The −213 to + 1090 region, containing the promoter, 5′-UTR, and CDS of *carD* and the promoter of *ascarD* on the antisense strand (abbreviated as PUCP), was translationally fused to *lacZ* to construct the PUCP plasmid (*Figure 5A*), in which the expression of the *carD-lacZ* chimeric transcript was expected to be regulated by the *cis*-encoded AscarD. However, in the PUCP$_{mut}$ plasmid, the −10 motif of *ascarD* is mutated (GGGTAC is mutated to GGGCGC) and could not transcribe AscarD, so the expression of the *carD-lacZ* transcript will not be affected by this antisense RNA. We then transformed the two plasmids into mc$^2$155 cells and measured their β-galactosidase activities. As shown in *Figure 5B*, mycobacterial cells transformed with the PUCP$_{mut}$ plasmid exhibited higher β-galactosidase activity than those with the PUCP plasmid. This result indicates that AscarD repressed the expression of *carD-lacZ* transcript, and blocking the transcription of *ascarD* derepressed this regulation.

To further explore the regulatory role of AscarD on *carD* expression, we overexpressed *ascarD* on a multiple-copy plasmid to construct *ascarD* high-expressing strain (*ascarD*$_{OE}$) and knockdown the transcription of *ascarD* to construct the *ascarD* low-expressing strain (*ascarD*$_{KD}$, *Figure 5—figure supplement 1A, B*) and examined the changes of CarD protein levels in these strains. As shown in *Figure 5C*, compared to the control strain, the CarD level in the *ascarD*$_{OE}$ strain was reduced, while in the *ascarD*$_{KD}$ strain it was significantly increased. This result indicates that AscarD inhibits the synthesis of CarD, which is consistent with the *lacZ* reporter assay data described above.

In addition, previous studies showed that CarD-impaired mycobacterial cells are more sensitive to oxidative stress, DNA damage, and the effect of some antibiotics (*Garner et al., 2014*; *Stallings et al., 2009*; *Weiss et al., 2012*). To further investigate the effect of AscarD on CarD expression and its biological function, we examined the tolerance of AscarD overexpression strain to the above-mentioned stresses. The results showed that overexpression of AscarD significantly enhanced the sensitivity of mycobacterial cells to these stresses (*Figure 5D*). This result is consistent with the experimental data described above, allowing us to conclude that AscarD, when fully induced, significantly inhibits the expression of CarD and affects its function.

It is important to point out that the inhibitory effect of antisense RNA on target genes can occur at the post-transcriptional level (reducing transcript stability) and/or the translational level (inhibiting the transcript translation) (*Georg and Hess, 2011*). To determine the inhibition mode, we measured the transcript level of *carD* in the *ascarD*$_{OE}$ strain. The *carD* transcript levels were significantly higher in the *ascarD*$_{OE}$ strain than those in the control strain (*Figure 5—figure supplement 1C*), illustrating that overexpressed AscarD increases, rather than decreases, the stability of *carD* transcripts. Since AscarD

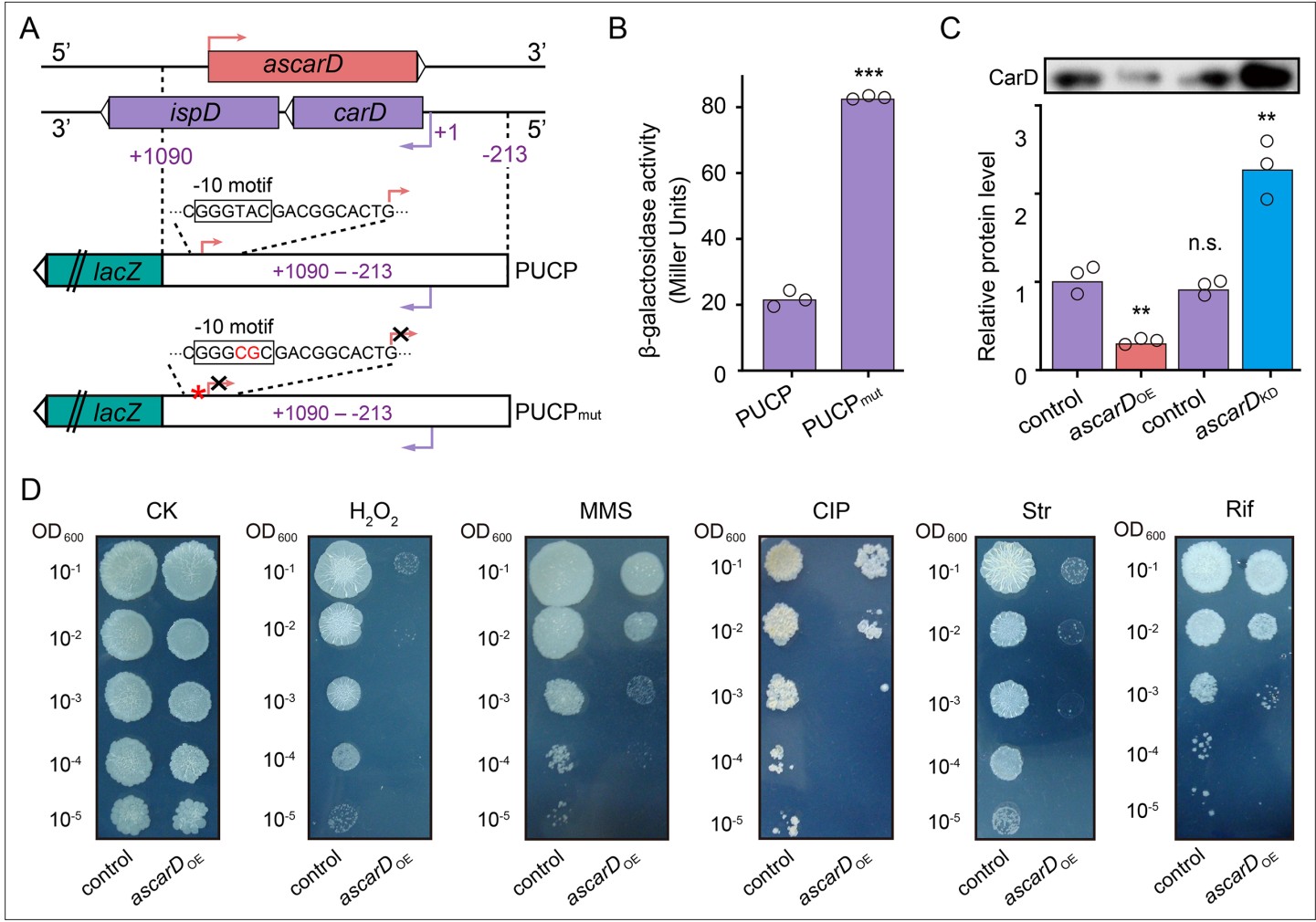

**Figure 5.** AscarD negatively regulates *carD*. (**A**) A schematic diagram of PUCP and PUCP_{mut} plasmids construction (see a detailed description in Experimental section). (**B**) β-Galactosidase activities of mc²155 strains transformed with PUCP or PUCP_{mut} plasmid. Individual data for the three biological replicates are shown in the corresponding columns. (**C**) CarD protein levels in different strains. Mycobacterial cells were harvested at the mid-stationary phase (MSP). The upper part shows the Western blot with CarD levels, and the histogram below it shows the quantitative statistics of Western blot results. Statistical test was done using the Student's t-test, with ** indicating p-value <0.01, *** indicating p-value <0.001, and n.s. indicating p-value >0.05. (**D**) The tolerance of *ascarD*_{OE} and control strains to oxidative stress, DNA damage, and antibiotic stimulation, respectively. Serially diluted bacterial suspensions were separately spotted onto normal 7H10 plate (CK) or plates containing 0.3 mM $H_2O_2$, 0.05% methanesulfonate (MMS), 0.2 μg/ml of ciprofloxacin (CIP), 0.1 μg/ml of streptomycin (Str), or 5 μg/ml of rifamycin (Rif), respectively.

The online version of this article includes the following source data and figure supplement(s) for figure 5:

**Source data 1.** β-Galactosidase activities of mc²155 strains transformed with PUCP or PUCP_{mut} plasmid (numerical data for *Figure 5B*).

**Source data 2.** CarD protein levels in different strains (numerical data for *Figure 5C*).

**Figure supplement 1.** The expression levels of *carD* and *ascarD* in different strains.

only reduces the CarD protein level, but not the transcript level, we speculated that AscarD inhibits *carD* expression at the translational level.

Under starvation conditions, AscarD was highly induced to inhibit CarD protein synthesis. Since CarD protein levels are not only reduced during nutrient starvation, but also reduced under hypoxic and acidic conditions, we wanted to know whether *ascarD* is also induced under such stress conditions. To address this question, we monitored the RNA level of AscarD under the two stress conditions by qRT-PCR. The results showed that the AscarD level increased by 4.5 and 2.3 times in response to hypoxia and acid stress, respectively. These data indicate that AscarD was upregulated in response to a variety of stimuli to inhibit the protein synthesis of CarD and help mycobacterial cells adapt to the stress environment.

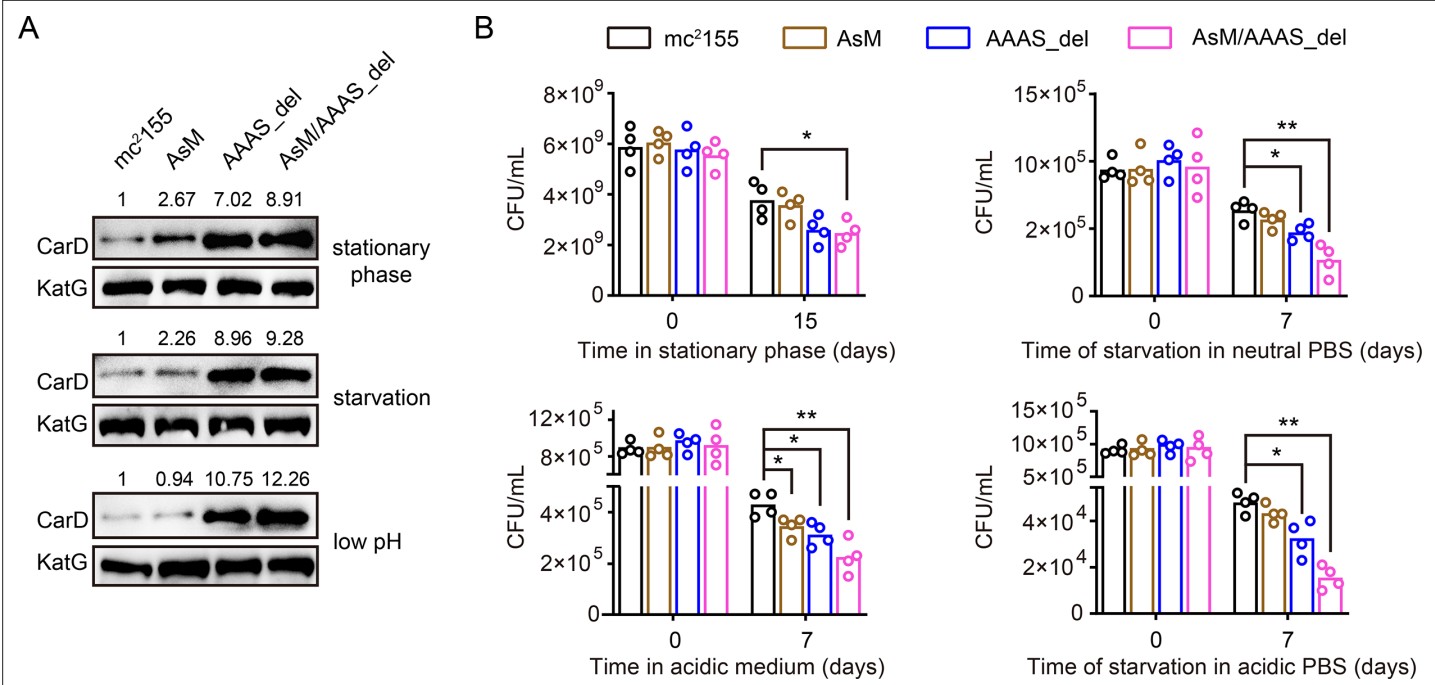

**Figure 6.** AscarD and Clp protease coregulate CarD-mediated mycobacterial adaptive response. (**A**) Changes in CarD protein levels of different strains under various stress conditions. AsM, AAAS_del, and AsM/AAAS_del represent, respectively, AscarD promoter mutant, AAAS motif deletion, and double mutant strains. (**B**) Survival of different mycobacterial cells under various stress conditions. Statistical test was done using the Student's $t$-test, with * indicating p value <0.05, and ** indicating p value <0.01.

The online version of this article includes the following source data for figure 6:

**Source data 1.** Survival of different mycobacterial cells under various stress conditions (numerical data for *Figure 6B*).

## AscarD and Clp protease coregulate CarD-mediated mycobacterial adaptive response

AscarD and Clp protease regulate CarD at different levels. To explore which of these two regulations is dominant and whether there is a synergistic effect between the two, we examined the changes in CarD levels and bacterial survival rates in different mutant strains. As mentioned earlier, deletion of the 'AAAS' motif blocked the degradation of CarD by Clp protease. To block the regulation of CarD by AscarD, we mutated the promoter of *ascarD* in *M. smegmatis* genome and constructed a mutant strain, referred as AsM. In addition, to block the regulation of CarD by both AscarD and Clp protease, we also constructed a double mutant strain AsM/AAAS_del with a mutation in the promoter of *ascarD* and the deletion of the 'AAAS' motif of CarD. Next, we investigated the changes in CarD levels and bacterial survival of these strains under stress conditions. The Western blot results showed that, compared to the wild-type strain, CarD levels in the AsM strain slightly increased, while CarD levels in the AAAS_del strain increased significantly (*Figure 6A*). This indicates that under these stress conditions tested, Clp protease dominates the regulation of CarD levels.

While bacterial survival assays showed that relieving the regulation of AscarD on CarD had a weak impact on the survival of mycobacterial cells, relieving the regulation of Clp protease on CarD had a moderate impact, and relieving both regulatory mechanisms strongly affected the survival of myco-bacterial cells (*Figure 6B*). These results indicate that AscarD and Clp protease are both important for the survival of mycobacterial cells under stress conditions. While Clp protease is responsible for the rapid reduction of CarD protein levels, AscarD further reduces CarD protein levels by inhibiting *carD* translation. Their combined action helps mycobacterial cells save energy in the stress conditions by preventing the futile cycle of CarD synthesis and its degradation by Clp protease. Moreover, AscarD could prevent mycobacterial cells from overaccumulating CarD in the absence of the expression of Clp protease, which is essential for their survival under stress conditions. Altogether, AscarD and

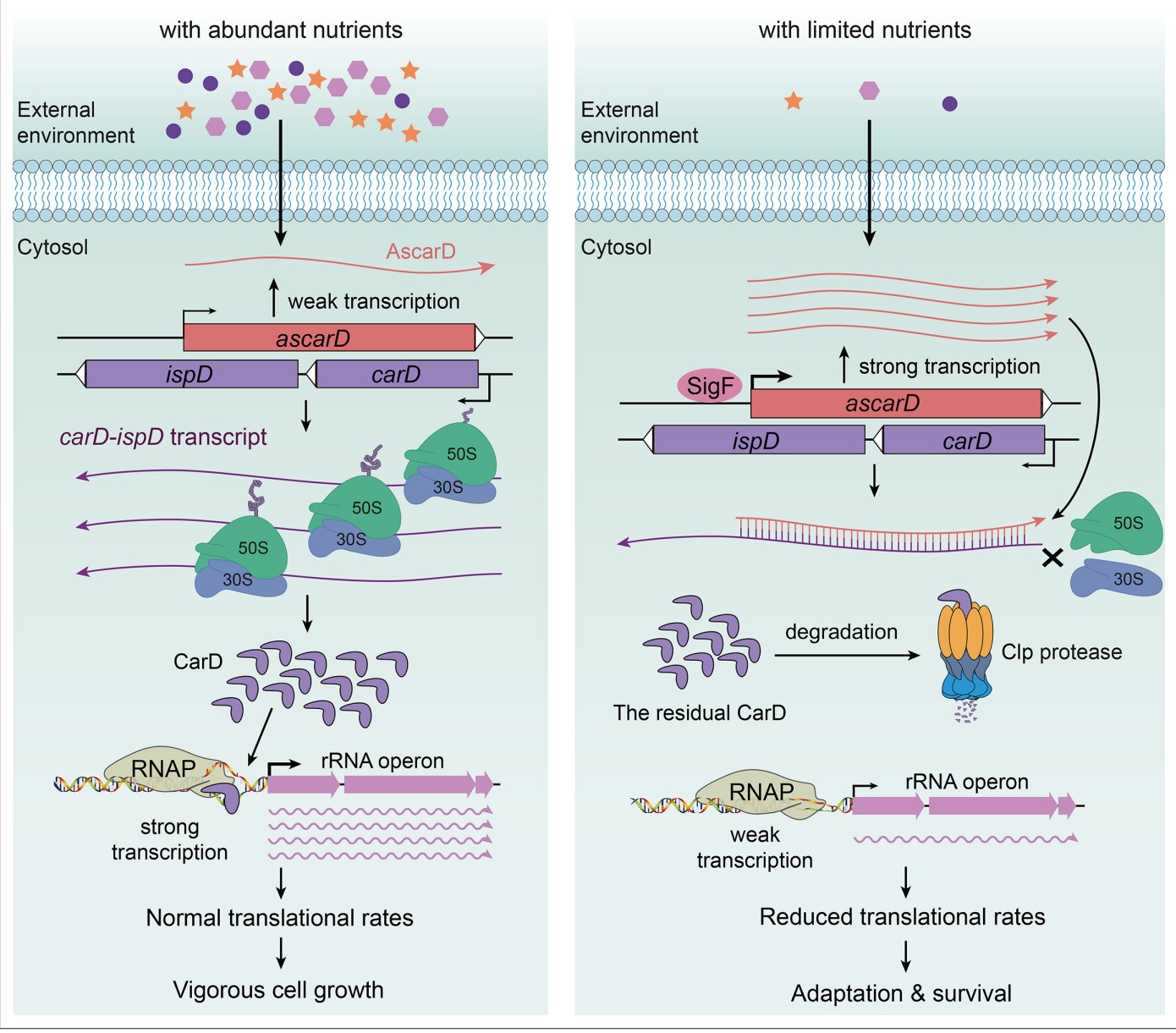

**Figure 7.** AscarD and Clp protease work together to regulate CarD-mediated starvation response. The left and right panels represent the mycobacterial cells under nutrient-rich and nutrient-starved conditions, respectively.

The online version of this article includes the following figure supplement(s) for figure 7:

**Figure supplement 1.** Alignment of mycobacterial *carD* promoter sequences.

Clp protease work synergistically to decrease the CarD level to help mycobacterial cells respond to various stresses.

## Discussion

In this paper, we present an in-depth study on the regulation of CarD expression and demonstrate that CarD is coregulated by AscarD antisense RNA and Clp protease under starvation conditions. Based on these results, along with those published by others, we propose a new mechanism for mycobacterial adaption to the starvation conditions, namely, that mycobacterial cells adjust their transcriptional and translational rates by regulating the CarD levels in response to the environmental conditions (*Figure 7*).

Under abundant nutrition, mycobacterial cells use CarD to stabilize RPo (*Bae et al., 2015*; *Davis et al., 2015*; *Rammohan et al., 2016*; *Rammohan et al., 2015*), promoting the transcription of rRNA and other related genes (*Garner et al., 2014*; *Rammohan et al., 2016*; *Srivastava et al., 2013*) to ensure vigorous cell growth (*Figure 7*, left). However, when external nutrition gets scarce, SigF-regulated expression of AscarD is induced and it hybridizes with *carD* mRNA to prevent the translation of CarD protein (*Figures 4 and 5*). Meanwhile, the residual CarD protein is effectively degraded by the Clp complex to keep CarD at a very low level (*Figures 1–3*), which potentially reduces the stability of RPo and diminishes the synthesis of rRNA; these processes combine to slow down the rate of transcription and translation in mycobacterial cells (*Figure 7*, right). When nutrients are available, AscarD transcription is inhibited and *carD* mRNA gets translated to resume the normal CarD level and ensure the regrowth of mycobacterial cells (*Figure 1F*). Overall, these findings contribute to a better understanding of the mechanisms of mycobacterial adaptation to starvation and provide certain clues that might help in the treatment of tuberculosis.

## Mycobacterial CarD defines a distinct adaptive response mechanism

Before this work, the best-known starvation response mechanism in *Mycobacterium* was the stringent response mediated by (p)ppGpp. Yet, its detailed mechanism is still not entirely clear today, although it is considered similar to the well-characterized stringent response mechanisms reported in *E. coli* and *B. subtilis* (*China et al., 2012*; *Prusa et al., 2018*; *Weiss and Stallings, 2013*). In *E. coli*, (p)ppGpp is synthesized in large quantities in response to starvation and directly interacts with RNAP to destabilize the RPo formed on rRNA genes and consequently reduces the rRNA synthesis (*Gourse et al., 2018*). However, (p)ppGpp in *B. subtilis* does not directly interact with RNAP but instead decreases the intracellular GTP content to destabilize the RPo formed on the genes that start from guanine, such as the rRNA genes (*Kriel et al., 2012*; *Tojo et al., 2010*). In mycobacteria, the exact effect and role of ppGpp on rRNA transcription are unclear, but (p)ppGpp likely inhibits the transcription of mycobacterial rRNA by affecting the stability of RPo (*Prusa et al., 2018*), which is similar to the CarD-mediated starvation response. Of course, these two mechanisms also have their unique features. First, (p)ppGpp reduces the stability of RPo (*China et al., 2012*; *Tare et al., 2013*), while CarD enhances its stability (*Bae et al., 2015*; *Davis et al., 2015*; *Rammohan et al., 2016*; *Rammohan et al., 2015*). Second, (p)ppGpp is rapidly synthesized in response to starvation, while CarD is effectively degraded under starvation conditions. Despite the differences between the two mechanisms, they basically work in the same way and ultimately help the mycobacterial cells adapt to starvation by reducing the rRNA synthesis. It should be noted that Stallings et al. previously reported that CarD is required for stringent response in *M. smegmatis* (*Stallings et al., 2009*). However, our data showed that CarD is effectively degraded under starvation conditions where stringent response usually occurs. This may seem to contradict the previous study, but since both CarD and (p)ppGpp interact with RNAP (*Gourse et al., 2018*; *Stallings et al., 2009*), the two molecules might have complex effects on RNAP that remain to be disentangled.

In addition, our study found that mycobacterial cells reduce CarD levels in response to hypoxic conditions. By analyzing the RNA-seq data for *M. tuberculosis* (*Zhu et al., 2019*), we found that 18 of the top 20 upregulated genes (*Supplementary file 1*) in CarD^K125A^ strain (a mutant with predicted weakened affinity of CarD to DNA) belong to the previously identified dormancy regulon (*Voskuil et al., 2003*), and 31 out of 48 dormancy regulon genes are significantly upregulated. These data indicate that CarD represses the expression of dormancy regulon genes, and the reduction of CarD level during starvation or hypoxia may derepress these genes and facilitate mycobacterial dormancy. Since pathogenic mycobacteria usually live in a nutrient-deprived and hypoxic environment after infecting the host, we believe that CarD plays an important role in the dormancy and persistence of pathogenic mycobacteria in the host cells. Taken together, the CarD-mediated mycobacterial adaptive response mechanism is multifaceted; the reduction of CarD not only downregulates the transcription of rRNA to help mycobacterial cells adapt to nutritional starvation, but also enhances the expression of dormancy regulon genes to help pathogenic mycobacteria entering into a dormant state.

## Efficient degradation of CarD during the stationary phase

The efficient degradation of CarD in the stationary phase may be caused by the increased expression of ClpC1. Notably, the increase in ClpC1 level during the stationary phase is also observed in *Mycobacterium avium* (*Enany et al., 2021*), and the content of ClpC or other AAA+ unfoldases (ClpA, ClpX,

etc.) in many bacteria also increases significantly during the stationary phase (*Chaussee et al., 2008*; *Cohen et al., 2006*; *Laakso et al., 2011*; *Michel et al., 2006*; *Sowell et al., 2008*). This indicates that upregulation of Clp protease may be a conserved regulatory mechanism for bacteria to cope with starvation stress. Additionally, *Schmitz and Sauer, 2014* previously suggested that binding of an AAA$^+$ unfoldase strongly stimulates the peptidase activity of ClpP1P2 and stabilizes the conformation of the active complex. Therefore, the increased ClpC1 level during the stationary phase not only accelerates the unfolding of CarD, but also enhances the proteolytic activity of ClpP1P2, which ultimately mediates the effective degradation of CarD.

Considering the complexity of intracellular regulation, we speculate that there may be other reasons for the efficient degradation of CarD during the stationary phase. First, in the exponential phase, CarD may be protected by a certain protein complex. Garner et al. previously suggested that CarD–RNAP interaction protects CarD from proteolytic degradation (*Garner et al., 2017*). Therefore, RNAP (or other proteins) may protect CarD in the exponential phase. Then, after the mycobacterial cells enter the stationary phase, CarD would be detached from RNAP (or other proteins) through unknown mechanisms and be effectively degraded by Clp protease. Second, besides Clp protease, degradation of CarD may require an adaptor protein. For example, in *Caulobacter crescentus*, CpdR directly controls PdeA degradation by acting as a phosphorylation-dependent adaptor protein for the ClpXP protease (*Abel et al., 2011*). We speculate that there is possibly an adaptor protein that recognizes CarD under the starvation condition and delivers it to the Clp protease for degradation. Third, degradation of CarD by Clp protease may be affected by its modification. For example, certain protein substrates in *B. subtilis* are degraded by Clp protease only after their arginine residues are phosphorylated (*Trentini et al., 2016*). CarD might undergo a similar structural modification under the starvation condition, which is specifically recognized and degraded by Clp protease.

## Role of AscarD in inhibition of the synthesis of CarD protein

The inhibitory effect of antisense RNAs on target genes generally occurs at the post-transcriptional level and/or the translational level (*Georg and Hess, 2011*). At the translation level, antisense RNAs mainly regulate the initiation of translation by blocking the SD sequence or adjacent regions of the target mRNA (*Georg and Hess, 2018*; *Saberi et al., 2016*; *Sesto et al., 2013*). In this study, we found that AscarD inhibited the synthesis of CarD protein but at the same time increased the stability of *carD* mRNA. Therefore, we speculated that the inhibition of CarD protein synthesis by AscarD is likely to occur at the translation level. So how does AscarD inhibit *carD* mRNA translation? Does its 3′-end cover the SD sequence of *carD* mRNA? It should be noted that we failed to identify the 3′-end of AscarD through 3′-RACE, but some of our results showed that AscarD does extend to the region that blocks the SD sequence of *carD* mRNA. We think this may be the main way that AscarD affects CarD protein synthesis. Of course, in addition to inhibiting the translation of *carD* mRNA, AscarD may also affect the synthesis of CarD protein in other ways. For example, transcription and translation in mycobacteria appear to be coupled (*Johnson et al., 2020*), such that the lead ribosome potentially contacts RNAP and forms a supramolecular complex. Therefore, a head-on RNAP on the antisense strand may become an obstacle to the RNAP on the sense strand and the trailing ribosomes, which may affect the synthesis of CarD protein.

Clp protease degrades CarD at the post-translational level, while AscarD inhibits CarD synthesis at the translational level. This two-tier mechanism allows mycobacterial cells to tightly control the CarD level. For example, when the content of Clp is insufficient or its function is lost, CarD may not be efficiently degraded; in that case, AscarD could prevent overaccumulation of CarD by inhibiting its synthesis. In fact, Clp protease is responsible for degrading unfolded/misfolded proteins that accumulate during stress conditions (*LaBreck et al., 2017*) and contributes to the clearance of truncated peptides from stalled ribosomes (*Gottesman et al., 1998*). The amount of these 'competitive substrates' increases under stress, for example, at high temperatures (*Fujihara et al., 2002*), which may result in the insufficient degradation of CarD by Clp protease. Furthermore, some natural compounds have been reported to inhibit the activity of Clp protease (*Moreno-Cinos et al., 2019*; *Raju et al., 2012*), suggesting that mycobacterial cells may face reduced or lost activity of Clp during in vitro growth or after infection of the host. In such situations, AscarD would be particularly important. Additionally, the presence of AscarD also helps mycobacterial cells save energy by preventing the futile cycle of CarD synthesis in the starvation condition and its degradation by Clp protease, which

may be harmful to mycobacterial survival. Taken together, our data show that AscarD works together with Clp protease to maintain CarD at the minimal level to help mycobacterial cells cope with the nutritional stress.

## Regulation of *carD* at the transcriptional level

Previous reports showed that the transcription of *carD* is regulated by SigB, but *carD* can still be effectively transcribed in the *sigB* knockout strain (*Hurst-Hess et al., 2019*). Since the −10 elements recognized by SigA and SigB are somewhat similar in mycobacteria, we speculate that SigA and SigB jointly regulate the *carD* expression, with SigA as the primary $\sigma$-factor responsible for the basal transcription of *carD*, and SigB as an alternative $\sigma$-factor responsible for the stimulated transcription of *carD* under stress conditions, which may also be the reason for the increasing *carD* expression after treatment with DNA-damaging agents (*Figure 1A, B*).

In addition, in *Rhodobacter*, CarD negatively regulates its own promoter, and the negative effect mainly depends on the extended −10 element (TGN) and the adjacent spacer of the promoter (*Henry et al., 2021*). At present, it is unclear whether mycobacterial CarD is autoregulated. After analyzing *carD* promoters from 91 different mycobacterial species, we found that mycobacterial *carD* also contains a conserved extended −10 element (*Figure 7—figure supplement 1*). Considering that only a few 'TANNNT' motifs in mycobacteria are preceded with extended −10 element (*Cortes et al., 2013*; *Henry et al., 2020*), we speculate that the highly conserved extended −10 element in the *carD* promoter may play an important role in the maintaining and regulating its basal activity. Moreover, a specific feature (T-rich) in the spacer immediately upstream of the extended −10 element contributes greatly to the autoregulation of *Rhodobacter* CarD. In *Mycobacterium*, there is no similar spacer, but there is a highly conserved dinucleotide 'CG' immediately upstream of the extended −10 element (*Figure 7—figure supplement 1*). Based on this limited information, it is difficult to determine whether mycobacterial CarD is autoregulated. In addition, it is worth mentioning that there are two conserved regions upstream of the *carD* core promoter regions (*Figure 7—figure supplement 1*). We speculate that these sequences may play a role in regulating the expression of *carD*, but to our knowledge, no potential transcription factor that can bind to these two sequences has been identified. Future studies to explore the function of these conserved elements will help to fully elucidate the regulatory mechanism of CarD.

# Materials and methods

## Bacterial strains and growth condition

*E. coli* strains were cultivated in lysogeny broth medium at 37°C. *M. smegmatis* mc$^2$155 wild-type strain (*Yang et al., 2012*) and its derivatives were grown at 37°C in Middlebrook 7H9 medium supplemented with 0.5% (vol/vol) glycerol and 0.05% (vol/vol) Tween 80, or on Middlebrook 7H10 agar supplemented with 0.5% (vol/vol) glycerol. *M. bovis* BCG and *M. tuberculosis* H37Ra strains (*Yang et al., 2018*) were grown at 37°C in 7H9 medium supplemented with 0.5% glycerol, 0.05% Tween 80% and 10% OADC (oleic acid, albumin, dextrose, and catalase), or on Middlebrook 7H11 agar supplemented with 0.5% glycerol and 10% OADC. When required, antibiotics were added at the following concentrations: kanamycin (Kan), 25 µg/ml; hygromycin (Hyg), 50 µg/ml; streptomycin (Str), 10 µg/ml. The strains used in this study are listed in *Supplementary file 2*.

Hartmans–de Bont (HDB) minimal medium, prepared according to reference (*Smeulders et al., 1999*), was used for starvation experiments. Briefly, 1 l of HDB medium contained: 10 mg of EDTA, 100 mg of MgCl$_2$·6H$_2$O, 1 mg of CaCl$_2$·2H$_2$O, 0.2 mg of NaMoO$_4$·2H$_2$O, 0.4 mg of CoCl$_2$·6H$_2$O, 1 mg of MnCl$_2$·2H$_2$O, 2 mg of ZnSO$_4$·7H$_2$O, 5 mg of FeSO$_4$·7H$_2$O, 0.2 mg of CuSO$_4$·5H$_2$O, 1.55 g of K$_2$HPO$_4$, 0.85 g of NaH$_2$PO$_4$, 2.0 g of (NH$_4$)$_2$SO$_4$, 0.2% glycerol (vol/vol), and 0.05% Tween 80 (vol/vol). For the carbon starvation experiment, glycerol was removed; for the nitrogen starvation experiment, (NH$_4$)$_2$SO$_4$ was removed; for the phosphorus-starvation experiment, both K$_2$HPO$_4$ and NaH$_2$PO$_4$ were removed, while 50 mM 3-(*N*-morpholino) propanesulfonic acid was added to replace lost buffering capacity.

## Stimulation and starvation experiments

Mycobacterial cells were first grown to MEP in the normal 7H9 medium. For genotoxic reagent stimulation experiments, 10 mM H$_2$O$_2$, 0.1% MMS, and 10 µg/ml of CIP were separately added to the

MEP mc$^2$155 culture and maintained in roller bottle culture for additional 4 hr. For the PBS starvation experiment, the MEP mc$^2$155 cells were harvested, resuspended in PBS supplemented with 0.05% Tween 80, and maintained in roller bottle culture for 4 hr. For the carbon-, nitrogen-, and phosphorus-starvation experiments, harvested MEP cells were resuspended in the HDB medium with carbon, nitrogen, or phosphorus removed, respectively, and maintained in roller bottle culture for 4 hr for the mc$^2$155 cells, or 24 hr for the BCG and H37Ra cells. For the nutrient supplemented experiments, the abovementioned starved cultures were supplemented with the corresponding nutrients and maintained in roller bottle culture for additional 4 hr for the mc$^2$155 cells, or 24 hr for the BCG and H37Ra cells. For the acid stimulation experiments, the harvested MEP cells were resuspended in the HDB medium with a low pH value (pH 4.5) and maintained in roller bottle culture for 4 hr for the mc$^2$155 cells, or 24 hr for the BCG and H37Ra cells.

For anaerobic experiments, the modified Wayne model (*Wayne and Hayes, 1996*) was used. Briefly, 150 ml standard serum bottles containing 100 ml of 7H9 medium were used, in which methylene blue was added to the final concentration of 2 µg/ml to indicate oxygen content. The harvested MEP cells were reinoculated into the above serum bottles to make the final OD$_{600}$ of 0.02. Then, the serum bottles were sealed with butyl rubber stoppers, closed tightly with screwcaps, and incubated at 37°C with shaking. The mc$^2$155 cells were harvested 10 hr after the blue color disappeared completely, and the BCG and H37Ra cells were harvested 48 hr after the blue color completely disappeared. For the reaeration experiments, the abovementioned anaerobic cultures were transferred to roller bottles and harvested after the mycobacterial cells regrow.

## RNA isolation, reverse transcription, and qRT-PCR

The total RNA was extracted by the TRIzol method using mycobacterial cells equivalent to 30 OD$_{600}$ (e.g., 30 ml of a culture with OD$_{600}$ of 1), as described previously (*Li et al., 2017*). The quality and concentration of total RNA were analyzed by NanoDrop 2000 (Thermo Scientific, USA). For reverse transcription, the total RNA was treated with DNase I (Takara Biotechnology, Japan) to remove any DNA contamination. The first-strand cDNA was synthesized using reverse transcriptase from the PrimeScript RT reagent kit (Takara Biotechnology, Japan) according to the manufacturer's instructions. The cDNA of *carD* or *ascarD* was synthesized using gene-specific primers RT-*carD*-R or RT-*ascarD*-R instead of random primers, which allowed us to distinguish between the two transcripts. For qRT-PCR, the reaction was performed in ABI 7500 (Applied Biosystems, USA) under the following conditions: 95°C for 10 s, 60°C for 10 s, and 72°C for 10 s for 40 cycles. Relative quantification of gene expression was performed by the $2^{-\Delta\Delta CT}$ method (*Livak and Schmittgen, 2001*). *sigA* was used as a reference gene for the determination of relative expression. The primers used in this study are listed in *Supplementary file 3*.

## Construction of the *ascarD* and *carD* overexpression strains

The overexpression plasmids of *ascarD* and *carD* were constructed based on the multicopy plasmid pMV261. For *ascarD* overexpression, we cloned the *ascarD* promoter and coding region into the pMV261 vector between the *Xba*I and *Hind*III restriction sites. For *carD* overexpression, we cloned the coding sequence into pMV261 between the *EcoR*I and *EcoR*V restriction sites, which allowed *carD* to be transcribed from the *hsp60* promoter on the vector. The overexpression plasmids were then transformed into mc$^2$155 cells to obtain the overexpression strains. The primers used are listed in *Supplementary file 3*.

## CRISPRi-mediated gene knockdown strategy

CRISPR/dCas9-mediated gene knockdown strategy (*Singh et al., 2016*) was carried out to construct the AscarD$_{KD}$ strain. Briefly, pRH2502 plasmid-containing *int* and *dcas9* genes was integrated into the mc$^2$155 genome to generate Ms/pRH2502 strain (*Supplementary file 2*). pRH2521 plasmid containing the small guide RNA (sgRNA) targeting *ascarD* was transformed into Ms/pRH2502 strain to obtain the final AscarD$_{KD}$ strain (*Supplementary file 2*). The expression of both *dcas9* and sgRNA requires the induction by ATc. The transcription of the target gene (*ascarD*) was inhibited with the induction by 50 ng/ml of ATc, and the inhibition efficiency was assessed by qRT-PCR. It is worth noting that dCas9:sgRNA complex exhibits a strong inhibitory effect on the expression of a gene after it combines with the coding strand of the gene, but almost does not affect the expression of the gene

when it combines with the template strand. The sgRNA we designed is combined with the coding strand of *ascarD* (i.e., the template strand of *ispD*), so it has a strong inhibitory effect on the transcription of *ascarD* (reduced 18.8 ± 2.6 times as quantitated by qRT-PCR) but has almost no effect on the transcription of *ispD*. The inhibition efficiency is shown in **Figure 5—figure supplement 1A, B**, and all related primers are listed in **Supplementary file 3**.

## CRISPR/Cpf1-mediated mutagenesis

CRISPR/Cpf1-mediated mutagenesis was carried out as described previously (**Yan et al., 2017**). For *clpP2* conditional mutant (*clpP2*CM) construction, an exogenous *clpP2* gene amplified with *clpP2*-F/R primer pair (**Supplementary file 3**) was ligated to the pRH2502 integration plasmid to obtain pRH2502-*clpP2* recombinant plasmid, in which *clpP2* is under the control of ATc-inducible promoter $P_{UV15tetO}$. The pRH2502-*clpP2* plasmid was then transformed and integrated into the mc²155 genome by *attB–attP*-mediated site-specific recombination, to obtain Ms/pRH2502-*clpP2* strain. Finally, the endogenous *clpP2* gene on Ms/pRH2502-*clpP2* genome was mutated (pretranslational termination) using the CRISPR/Cpf1-mediated mutagenesis. Thus, *clpP2* could be expressed normally in the *clpP2*CM strain only upon the addition of 50 ng/ml ATc, but could not do so when ATc was absent. It should be noted that, in principle, the *clpP2*CM strain cannot grow in the 7H9 medium without ATc. But when we first cultivated the *clpP2*CM cells to exponential phase in ATc-containing 7H9 medium, then harvested the cells, washed them, and inoculated them into ATc-free 7H9 medium, *clpP2*CM cells can grow slowly.

## β-Galactosidase experiment

For PUCP construction, the −213 to +1090 region of *carD*, containing the *carD* promoter, 5′-UTR, *carD* CDS, and *ascarD* promoter on the antisense strand, was translationally fused to *lacZ*; for PUCP$_{mut}$ construction, the modified −213 to +1090 region of *carD*, containing the *carD* promoter, 5′-UTR, *carD* CDS, and the mutated *ascarD* promoter (GGGTAC was mutated to GGGCGC) on the antisense strand, was translationally fused to *lacZ*. Then the two plasmids were transformed into the mc²155 strain to measure the β-galactosidase activity. The detailed steps for β-galactosidase activity determination were carried out according to references (**Ali et al., 2017**; **Tang et al., 2014**).

## 5′-Rapid amplification of cDNA ends

To identify the TSS of *ascarD*, 5′-RACE analysis was performed with RNA extracted from mc²155 cells at midstationary phase grown in 7H9 medium. The 5′-RACE experiment was performed as described previously (**Zaunbrecher et al., 2009**). The primers used are listed in **Supplementary file 3**.

## Western blot

For internal reference in the Western blot experiments, we used SigA or KatG as indicated. In the stress stimulation experiments (**Figure 1B**), SigA was used as an internal control because its level is not affected by the test stimuli. However, the SigA protein level in the stationary phase is significantly lower than that in the log phase (**Gottesman et al., 1998**), so when we studied the protein levels in several growth phases, SigA was not used as an internal control. After many tests, we found that the protein level of KatG remained basically unchanged throughout the growth stage, so KatG was used as an internal control in those experiments. (Note: KatG is highly induced under oxidative stress conditions, so it was not suitable for use as an internal control in the stress stimulation experiments.) CarD or KatG was detected using the CarD- or KatG-specific rabbit polyclonal antibodies prepared by Dia-An Biotech, Inc (Wuhan, China). For SigA detection, His × 6 tag was fused to the C-terminus of SigA by inserting its coding sequence immediately upstream of the *sigA* stop codon in the mc²155 genome, and the modified SigA-His × 6 protein was detected using rabbit polyclonal antibody to His × 6 (Yeasen Biotech Co., Shanghai, China). For Western blot assays, the amount of total protein loaded in each lane was the same, and the detailed procedures were as described previously (**Hnasko and Hnasko, 2015**). For quantification of Western blot results, Image J software was used. The intensities of bands in each lane were individually measured, and the intensities of the target protein were normalized with respect to their corresponding loading control. In each panel, the normalized value of the first sample was set to 1, and the values of other samples were represented by the fold changes of their normalized value relative to the first sample.

## Pull-down assay

The His × 6 tagged ClpC1 (ClpC1-His) and ClpX (ClpX-His) recombinant proteins were expressed and purified from *E. coli* BL21(DE3). After purification, the eluate containing ClpC1-His/ClpX-His protein was dialyzed overnight at 4°C, then incubated with 1 mM ATP for 1 hr before loading onto the Ni-NTA resin. For pull-down assay, the resin-bound ClpC1-His or ClpX-His protein was separately incubated with lysate extracted from exponential mc$^2$155 cells at room temperature for 30 min. The resins were washed with 50 mM imidazole for five times and eluted with 500 mM imidazole. The eluents were subjected to immunoblot assay using the antibodies indicated.

## Bacterial survival assay

For the stationary-phase survival assay, mycobacterial cells were first grown to the stationary phase, followed by keeping them at 4°C, and the bacterial counts were performed on the 0th and 15th days thereafter. For stress survival assays, mycobacterial cells were first grown to the early-exponential phase (OD$_{600}$≈ 0.5) and then diluted 50-fold into acidic 7H9 medium (pH = 4.5), neutral PBS (pH = 7.0), or acidic PBS (pH = 4.5). The dilutions were kept at 4°C, and the bacterial counts were performed 0th and 7th days thereafter. For genotoxic stress survival assay, *ascarD*$_{OE}$ and control cells were grown to MEP (OD$_{600}$≈1.0), followed by diluting 10$^1$, 10$^2$, 10$^3$, 10$^4$, and 10$^5$ folds, respectively. Afterward, 3 µl of the bacterial suspension at each dilution level were separately spotted onto 7H10 plates containing either 0.3 mM H$_2$O$_2$, 0.05% MMS, 0.2 µg/ml of CIP, 0.15 µg/ml of Str, or 5 µg/ml of Rif, respectively. The plates were cultivated at 37°C for three days.

## Statistical analysis

Statistical testing was done using the Student's *t*-test (two-tailed), with *** indicating p value <0.001, ** indicating p value <0.01, * indicating p value <0.05, and n.s. indicating p value >0.05. Error bars indicate standard deviation of three biological replicates. (Biological replicates represent tests performed on different biological samples representing an identical time or treatment dose, while technical replicates represent multiple tests on the same sample.)

# Additional information

### Funding

| Funder | Grant reference number | Author |
|---|---|---|
| National Natural Science Foundation of China | 31900057 | Qing Tang |
| National Natural Science Foundation of China | 32171424 | Jin He |
| China Postdoctoral Science Foundation | 2019M662654 | Xinfeng Li |
| National Institutes of Health | | Michael Y Galperin |

The funders had no role in study design, data collection, and interpretation, or the decision to submit the work for publication.

### Author contributions

Xinfeng Li, Fang Chen, Conceptualization, Investigation, Methodology, Visualization, Writing - original draft; Xiaoyu Liu, Jinfeng Xiao, Investigation, Methodology, Visualization; Binda T Andongma, Qing Tang, Xiaojian Cao, Formal analysis, Methodology; Shan-Ho Chou, Michael Y Galperin, Visualization, Writing - original draft, Writing - review and editing; Jin He, Conceptualization, Funding acquisition, Project administration, Supervision, Writing - review and editing

### Author ORCIDs

Fang Chen http://orcid.org/0000-0003-1663-3737
Michael Y Galperin http://orcid.org/0000-0002-2265-5572

Jin He http://orcid.org/0000-0002-1456-8284

**Decision letter and Author response**
Decision letter https://doi.org/10.7554/eLife.73347.sa1
Author response https://doi.org/10.7554/eLife.73347.sa2

## Additional files

### Supplementary files
- Supplementary file 1. The top 20 upregulated genes in the CarD$^{K125A}$ mutant.
- Supplementary file 2. Strains used in this study.
- Supplementary file 3. Oligonucleotides used in this study.
- Transparent reporting form
- Source data 1. Raw images of the gels/blots presented in this article.

### Data availability
All data generated or analysed during this study are included in the manuscript and supporting file; Source Data files have been provided for Figures 1, 4, 5 and 6. These Source Data contain the numerical data used to generate the figures.

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
