## [Editor Report]

CarD is an RNA polymerase interacting protein that is essential for mycobacterial viability, the levels of which are important for controlling gene expression in mycobacteria during various stress conditions. This study reports two mechanisms that regulate levels of CarD under stress conditions, including starvation. The authors report that CarD levels are tightly regulated and that there was a dramatic decrease in the levels of CarD when cells switched from the nutrient-rich to the starvation condition. They discovered two synergistic mechanisms that led to this dramatic decrease in CarD. The first is SigF-dependent induction of antisense RNA of CarD (AscarD), which inhibits CarD translation and a second mechanism involving Clp protease-mediated degradation of intracellular CarD. The work will be of interest to researchers studying non-coding RNAs, microbial gene expression, physiology and stress response.

---

## [Decision Letter]

**Decision letter after peer review:**

Thank you for submitting your article "Clp protease and antisense RNA jointly regulate the global regulator CarD to mediate mycobacterial starvation response" for consideration by *eLife*. Your article has been reviewed by 3 peer reviewers, and the evaluation has been overseen by a Reviewing Editor and Bavesh Kana as the Senior Editor. The following individual involved in review of your submission has agreed to reveal their identity: Krishan Gopal Thakur (Reviewer #1).

Essential revisions:

1. From the study, it appears likely that SigF plays a key role in stress adaption by upregulating AscarD transcription. However, quantification of the expression of SigF under stress conditions is missing. Including data on SigF expression under starvation and other stress conditions will strengthen this study further.

2. Similarly, how transcript levels are being measured and normalized should be described where presented in the text/legend with perhaps more detail referenced in the Methods. As normalized data is provided, changes could technically be due either to increases of CarD transcript or decreases in the sigA transcript. Evidence or measurements that the sigA transcript doesn't change appreciably under these conditions needs to be provided. The same is technically true of the gel-based measurements of protein levels. If one assumes that the amount of material loaded in each lane is the same, then the data clearly shows what the authors suggest. However, this was not specifically mentioned anywhere leaving open the possibility that the over-expression of the control proteins results in an apparent repression of CarD. Please address this.

3. The authors have shown data that suggests that Clp proteases degrade CarD under stress conditions. CarD contains the "AAAS" sequence motif at the C-terminal region, which is recognized by Clp proteases. Mutating this sequence motif and quantifying mutant vs wild type CarD levels in cells under normal and stress conditions will further strengthen the claim that Clp proteases degrade CarD under stress conditions.

4. Although the paper presents a clear picture of two mechanisms (Proteolytic degradation and anti-sense) individually on CarD levels, the effects of these mechanisms on cellular phenotypes is less completely explored. For example, for the stationary phase dependent regulation of CarD, does this mechanism effect stationary phase survival? Similarly, what about survival/expression chages under the stress conditions examined? Some data is given, but more exploration of the cellular effects of these mechanisms would make the paper more appealing to a broad audience. With their ability of tune CarD levels via graded manipulation of asCarD or Clp, the authors could learn much more about the physiologic and global transcriptional effects of these regulatory mechanisms and this would be a major advance for the field. Relevant assays would include, (I) stationary phase survival assays and (I) RNA sequencing of cells with different levels of asCarD or Clp in different conditions to look at the global effects of these manipulations.

5. Although the individual effects of Clp and asCarD are documented, the paper would be stronger if it explored the relative importance and interaction of these two mechanisms in different conditions. Are they simply additive as the model figure suggests? Does the relative importance of each mechanism differ depending on the condition studied? Relevant experiments that could strengthen this point are: (1) explore conditions in which one mechanism is dominant (2) changing asCarD levels or ClpP in the same cell during different stress conditions to see if one mechanism is dominant.

Other comments:

1. In Figure 1B the transcript levels of CarD under ciprofloxacin and H2O2 appears comparable. However, there was about a three-fold increase in the levels of CarD in the presence of ciprofloxacin. Can authors comment on this?

2. It will also be interesting to see the temporal increase and decrease in the levels of CarD and AscarD transcripts during normal and stress conditions.

3. The statement in the paper that ClpP2 dependence can be interpreted directly as the involvement of P1P2 is only true if ClpP2 never functions alone. The authors essentially state that, but could this assumption be misleading?

4. The abbreviation PUPC should be explained.

5. A study in Rhodobacter sphaeroides suggests that CarD negatively regulates its own promoter (https://pubmed.ncbi.nlm.nih.gov/34152199/). Do the authors have any data on whether or not this is true in mycobacteria? What is known, if anything, about direct transcriptional control of the CarD promoter? In the absence of AscarD (which may physically interfere with CarD transcription), what happens to CarD transcript levels?

6. Can the authors comment on whether the proteolytic degradation of CarD will have an effect on the production of CarD in a feed-back loop?

7. RT-qPCR: No documentation of no RT reactions is given. This is important to exclude dna contamination.

8. Some justification of the use of KatG as a loading control in the western blots is needed, since this is unusual and subject to regulation under stress.

9. The rnaseq data presented in the discussion should be moved to results if it is new data. If it is previously published it can be discussed, but the table should be moved to Supplementary information.

10. A question that arises is whether CarD itself has any effect on the levels of AscarD or whether in published ChIP-seq data, CarD can be seen on the AscarD promoter. Do the authors have any data or comments here?

*Reviewer #1 (Recommendations for the authors):*

The manuscript is well written and easy to follow. The major conclusions of the study are well supported by the experimental data provided in the manuscript. However, I have the following comments and suggestions:

1. From the study, it appears likely that SigF plays a key role in stress adaption by upregulating AscarD transcription. However, quantification of the expression of SigF under stress conditions is missing in this study. I suggest including data on SigF expression under starvation and other stress conditions to strengthen this study further.

2. The authors have shown data that suggests that Clp proteases degrade CarD under stress conditions. CarD contains the "AAAS" sequence motif at the C-terminal region, which is recognized by Clp proteases. I suggest mutating this sequence motif and quantifying mutant vs wild type CarD levels in cells under normal and stress conditions. This will further strengthen the claim that Clp proteases degrade CarD under stress conditions.

3. In Figure 1B the transcript levels of CarD under ciprofloxacin and H2O2 appears comparable. However, there was about a three-fold increase in the levels of CarD in the presence of ciprofloxacin. Can authors comment on this?

4. It will also be interesting to see the temporal increase and decrease in the levels of CarD and AscarD transcripts during normal and stress conditions.

*Reviewer #2 (Recommendations for the authors):*

As I have indicated in the public review, I really enjoyed this paper and think that it should most certainly be published. I only have a few points of clarification and questions for the authors and they are listed below:

(1) How transcript levels are being measured and normalized should be described where presented in the text/legend with perhaps more detail referenced in the Methods. In particular, since normalized data is provided, changes could technically be due either to increases of CarD transcript or decreases in the sA transcript. Evidence or measurements that the sA transcript doesn't change appreciably under these conditions needs to be provided.

(2) The same is technically true of the gel based measurements of protein levels. If one assumes that the amount of material loaded in each lane is the same, then the data clearly shows what the authors suggest. However, I did not see this specifically mentioned anywhere leaving open the possibility that the over-expression of the control proteins results in an apparent repression of CarD.

(3) The statement in the paper that ClpP2 dependence can be interpreted directly as the involvement of P1P2 is only true if ClpP2 never functions alone. The authors essentially state that, but could this assumption be misleading?

(4) I did not see where the abbreviation PUPC was explained.

(5) One question I was left with was whether CarD itself has any effect on the levels of AscarD or whether in published ChIP-seq data, CarD can be seen on the AscarD promoter. Do the authors have any data or comments here?

(6) A study in Rhodobacter sphaeroides suggests that CarD negatively regulates its own promoter (https://pubmed.ncbi.nlm.nih.gov/34152199/). Do the authors have any data on whether or not this is true in mycobacteria? What is known, if anything, about direct transcriptional control of the CarD promoter? In the absence of AscarD (which may physically interfere with CarD transcription), what happens to CarD transcript levels?

Can the authors comment on whether the proteolytic degradation of CarD will have an effect on the production of CarD in a feed-back loop?

*Reviewer #3 (Recommendations for the authors):*

I think the paper could be stronger in two areas noted above, these are suggested experiments for each of the major points above in the public review.

Relevant experiments that could strengthen this area: stationary phase survival assays, RNA sequencing of cells with different levels of asCarD or Clp in different conditions to look at the global effects of these manipulations.

Although the individual effects of Clp and asCarD are documented, the paper would be stronger if it explored the relative importance and interaction of these two mechanisms in different conditions. Are they simply additive as the model figure suggests? Does the relative importance of each mechanism differ depending on the condition studied?

Relevant experiments that could strengthen this point: (1) explore conditions in which one mechanism if dominant (2) changing asCarD levels or ClpP in the same cell during different stress conditions to see if one mechanisms is dominant.

Technical comments:

RT-qPCR: No documentation of no RT reactions is given. This is important to exclude dna contamination.

Some justification of the use of KatG as a loading control in the western blots is needed, since this is unusual and subject to regulation under stress.

The rnaseq data presented in the discussion should be moved to results if it is new data. If it is previously published it can be discussed, but the table should be moved to SI.

---

## [Author Response]

Essential revisions:1. From the study, it appears likely that SigF plays a key role in stress adaption by upregulating AscarD transcription. However, quantification of the expression of SigF under stress conditions is missing. Including data on SigF expression under starvation and other stress conditions will strengthen this study further.

Thank you for this suggestion. We have measured the *sigF* transcript levels in *M. smegmatis* under different conditions and found that there is no significant induction of *sigF* expression under the various tested conditions including the stationary phase, which is similar to the previous report (Singh and Singh, 2008). However, previous studies have shown that SigF is active under stress conditions, especially in stationary phase. In this work, we also found that the SigF-dependent expression of AscarD is highly induced in the stationary phase. Moreover, based on the published transcriptome data of *M. smegmatis* (Li et al., 2017), we identified 39 genes with strict SigF recognition motif (GTTT-N(15-17)-GGGTA) in their promoter regions, and importantly, 37 of these genes are upregulated at least 5-fold in the stationary phase. All these cues suggest that SigF does function at stationary phase. As for the lack of coordination of SigF expression and function, we believe that it is caused by post-translational regulation. As reported, the σ factors are post-translationally regulated by a partner switching system, in which the anti-σ factors sequester the σ factor from the transcription machinery, and the anti-σ factor antagonists release the repression of σ factors. In *M. smegmatis*, the SigF partner switching system contains an anti-σ factor RsbW and several anti-σ factor antagonists (RsfA, RsfB, RsbW3, MSMEG_0586, MSMEG_5551, and MSMEG_6541) (Humpel et al., 2010; Oh et al., 2020). We have compared the transcript levels of these genes at different growth phases by analyzing our published transcriptome data (Li et al., 2017) and found that the expression of some anti-σ factor antagonists (especially RsbW3) is significantly upregulated at the mid-stationary phase (MSP) (see Author response table 1). Therefore, we conclude that under normal conditions, the function of SigF is inhibited by RsbW, and in response to stationary phase and/or environmental cues, the anti-SigF antagonists interact with RsbW in a negative manner, derepressing SigF and allowing it to interact with RNAP and express its regulon.

**Author response table 1. sa2table1:** Transcript levels of genes in the SigF partner switching system.

Gene	Locus tag	Function	RPKM value	Fold change(MSP/MEP)	
			*MEP*	MSP	
*sigF*	MSMEG_1804	RNAP σ factor F	463	573	1.24
*rsbW*	MSMEG_1803	anti-σ^F^	593	3,410	5.75
*rsbW3*	MSMEG_1787	anti-σ^F^ antagonist	17	6,055	356.18
*rsfA*	MSMEG_1786	anti-σ^F^ antagonist	128	473	3.69
*rsfB*	MSMEG_6127	anti-σ^F^ antagonist	1,764	298	0.17
*MSMEG_0586*	MSMEG_0586	Putative anti-σ^F^ antagonist	1,401	7,051	5.03
*MSMEG_5551*	MSMEG_5551	Putative anti-σ^F^ antagonist	7	45	6.85
*MSMEG_6541*	MSMEG_6541	Putative anti-σ^F^ antagonist	126	272	2.16

2. Similarly, how transcript levels are being measured and normalized should be described where presented in the text/legend with perhaps more detail referenced in the Methods. As normalized data is provided, changes could technically be due either to increases of CarD transcript or decreases in the sigA transcript. Evidence or measurements that the sigA transcript doesn't change appreciably under these conditions needs to be provided. The same is technically true of the gel-based measurements of protein levels. If one assumes that the amount of material loaded in each lane is the same, then the data clearly shows what the authors suggest. However, this was not specifically mentioned anywhere leaving open the possibility that the over-expression of the control proteins results in an apparent repression of CarD. Please address this.

Thank you for these suggestions. We have added a description of how the qRT-PCR data were measured and normalized in the relevant Figure legends and added a more detailed description in the Methods section. As for the transcript level of *sigA*, we used 16S rRNA as an internal reference and measured its expression under various conditions. The results showed that there was no appreciable change in the *sigA* transcript level under the tested conditions (see the Author response image 1). Finally, the amount of material loaded in each lane was the same. We have added this statement to the Methods section.

**Author response image 1. sa2fig1:** Relative transcript level of *sigA* under different conditions.

3. The authors have shown data that suggests that Clp proteases degrade CarD under stress conditions. CarD contains the "AAAS" sequence motif at the C-terminal region, which is recognized by Clp proteases. Mutating this sequence motif and quantifying mutant vs wild type CarD levels in cells under normal and stress conditions will further strengthen the claim that Clp proteases degrade CarD under stress conditions.

Thank you very much for this important suggestion. We have deleted the "AAAS" motif from the *M. smegmatis* CarD and measured CarD levels under stationary phase and starvation conditions. As shown in Author response image 2, in mc^2^155, CarD is almost completely degraded under stress conditions, while in the "AAAS" deletion mutant (AAAS_del), a large amount of CarD is still retained, meaning that the deletion of "AAAS" motif prevents the CarD degradation by Clp protease. These results further strengthen the notion that Clp protease degrades CarD under stress conditions. We have added this content to the manuscript.

**Author response image 2. sa2fig2:** Changes in CarD protein levels in different strains under (A) starvation condition or (B) stationary phase.

4. Although the paper presents a clear picture of two mechanisms (Proteolytic degradation and anti-sense) individually on CarD levels, the effects of these mechanisms on cellular phenotypes is less completely explored. For example, for the stationary phase dependent regulation of CarD, does this mechanism effect stationary phase survival? Similarly, what about survival/expression chages under the stress conditions examined? Some data is given, but more exploration of the cellular effects of these mechanisms would make the paper more appealing to a broad audience. With their ability of tune CarD levels via graded manipulation of asCarD or Clp, the authors could learn much more about the physiologic and global transcriptional effects of these regulatory mechanisms and this would be a major advance for the field. Relevant assays would include, (I) stationary phase survival assays and (I) RNA sequencing of cells with different levels of asCarD or Clp in different conditions to look at the global effects of these manipulations.

Yes, we agree that the effects of CarD on cellular phenotypes were not fully explored. To address this question, we have added some experiments. We mutated the *ascarD* promoter (AsM) in the genome, which relieved AscarD’s regulation of CarD; we also knocked out the AAAS motif of CarD in the genome (AAAS_del), which relieved the degradation of CarD by Clp protease. In addition, we also constructed a double deletion strain with *ascarD* promoter and CarD AAAS motif (AsM/AAAS_del), which relieved the regulation of CarD by both AscarD and Clp protease. Next, we investigated the survival of these strains under stationary phase and stress conditions. The results showed that relieving the AscarD regulation of CarD has a weaker impact on the survival of mycobacterial cells, while relieving the regulation of Clp protease of CarD has a moderate impact on the survival of mycobacterial cells. However, simultaneous removal of the two regulatory mechanisms strongly affects the survival of mycobacterial cells (see Author response image 3). These results suggest that AscarD and Clp protease are both important for the survival of mycobacterial cells under stress conditions, and they work synergistically to maintain CarD at a minimal level to help mycobacterial cells respond to various stresses.

**Author response image 3. sa2fig3:** The survival rate of different mycobacterial cells under various stress conditions. AsM, AAAS_del, and AsM/AAAS_del respectively represent AscarD promoter mutant, AAAS motif deletion, and double mutant strains.

As for RNA sequencing experiment, we carried out RNA-seq on an AscarD overexpression strain and a *clpP2* conditional mutant strain. RNA-seq analysis of the AscarD overexpression strain and its control strain revealed 482 differentially expressed genes (DEGs; Fold change>2, FDR<0.05), whereas RNA-seq of the *clpP2* conditional mutant strain and its control strain revealed 2487 DEGs (Fold change>2, FDR<0.05). These high numbers are not surprising, as both CarD and ClpP2 are global regulators in mycobacteria. We also classified the affected genes according to their involvement in major cellular pathways (using the KEGG pathway tool), but again, both overexpression of AscarD and depletion of ClpP2 were shown to affect multiple pathways in mycobacteria. It should be noted that the main goal of this work is to prove the regulation of CarD by anti-*carD* antisense RNA and Clp protease, and we believe we have proved that. Considering that a brief analysis of the RNA-seq data may not contribute much to the main conclusions, and an in-depth study of RNA-seq data may distract attention from the subject of this manuscript, we intend to report this result in future communications.

5. Although the individual effects of Clp and asCarD are documented, the paper would be stronger if it explored the relative importance and interaction of these two mechanisms in different conditions. Are they simply additive as the model figure suggests? Does the relative importance of each mechanism differ depending on the condition studied? Relevant experiments that could strengthen this point are: (1) explore conditions in which one mechanism is dominant (2) changing asCarD levels or ClpP in the same cell during different stress conditions to see if one mechanism is dominant.

According to your suggestion, we have investigated the changes of CarD levels of different strains (including mc^2^155, AsM, AAAS_del, and AsM/AAAS) under stationary phase and stress conditions. The results showed that, compared to the wild-type strain, the level of CarD in the AsM (AscarD promoter mutation) strain slightly increased, while the level of CarD in the AAAS_del (CarD “AAAS” motif deletion) strain increased significantly (see Author response image 4). This indicates that under the stress conditions tested, Clp protease dominates the regulation of CarD levels. Combined with the bacterial survival assays, we suggest that Clp protease is responsible for the rapid reduction of CarD protein levels, while AscarD further reduces CarD protein levels by inhibiting *carD* translation. Their combined action helps mycobacterial cells save energy in the stress conditions by preventing the futile cycle of CarD synthesis and its degradation by Clp protease. Moreover, AscarD could prevent mycobacterial cells from over-accumulating CarD in the absence of the expression of Clp protease, which is essential for their survival under stress conditions. Altogether, AscarD and Clp protease work synergistically to decrease the CarD level to help mycobacterial cells respond to various stresses.

**Author response image 4. sa2fig4:** Changes in CarD protein levels of different strains under various stress conditions. AsM, AAAS_del, and AsM/AAAS_del represent, respectively, AscarD promoter mutant, AAAS motif deletion, and the double mutant strains.

Other comments:1. In Figure 1B the transcript levels of CarD under ciprofloxacin and H2O2 appears comparable. However, there was about a three-fold increase in the levels of CarD in the presence of ciprofloxacin. Can authors comment on this?

The changes in *carD* transcript levels in the original Figure 1A were measured after 4 hours of stimulation. Considering that some stimulation conditions may have instantaneous effects on gene expression levels, we have shortened the stimulation time. The changes in *carD* transcript levels were therefore measured at 0.5 hour and 1 hour after stimulation. The results show that *carD* transcript levels were upregulated to varying degrees under these stimuli, which is similar to the results presented in the original Figure 1A. After 1 hour of ciprofloxacin stimulation, the transcription level of *carD* was upregulated ~4.3 times, which is significantly higher than the degree of *carD* induction by several other stimulation conditions. We believe that such a significant increase in CarD protein levels is due mainly to the increase in its transcript level. In addition, ciprofloxacin may also regulate the CarD protein levels through other unknown mechanisms (such as affecting the degradation of CarD by Clp protease). The specific mechanisms of protein level changes need to be further addressed. We have now replaced Figure 1A with the new data.

2. It will also be interesting to see the temporal increase and decrease in the levels of CarD and AscarD transcripts during normal and stress conditions.

Following your suggestion, we have measured the transcript levels of *carD* and AscarD at 0.5 h and 1 h after stress stimulation. The results showed that the *carD* transcript level increased to varying degrees under these stress conditions (see Author response image 5), and the AscarD transcript level only increased significantly under starvation conditions (see Author response image 5). We have replaced the original *carD* transcript data in Figure 1A with the new data.

**Author response image 5. sa2fig5:** Changes of *carD* or AscarD transcript levels under various stress conditions.

3. The statement in the paper that ClpP2 dependence can be interpreted directly as the involvement of P1P2 is only true if ClpP2 never functions alone. The authors essentially state that, but could this assumption be misleading?

Thank you for your comment. To be more rigorous, we have revised this statement. Where appropriate, we have modified ClpP1P2 to ClpP2 or Clp protease.

4. The abbreviation PUPC should be explained.

The abbreviation PUCP indicates the promoter, 5’-UTR, and CDS of *carD* and promoter of *ascarD* on the antisense strand. We have added this explanation to the manuscript.

5. A study in Rhodobacter sphaeroides suggests that CarD negatively regulates its own promoter (https://pubmed.ncbi.nlm.nih.gov/34152199/). Do the authors have any data on whether or not this is true in mycobacteria? What is known, if anything, about direct transcriptional control of the CarD promoter? In the absence of AscarD (which may physically interfere with CarD transcription), what happens to CarD transcript levels?

In *Rhodobacter sphaeroides*, CarD negatively regulates its own promoter, and the negative effect mainly depends on the extended -10 element (TGN) and the adjacent spacer of the promoter, which are highly conserved in the *Rhodobacter carD* promoters. To explore whether mycobacterial *carD* also auto-regulates, we analyzed *carD* promoters from 91 different mycobacterial species. The results showed that all analyzed *carD* promoters contain a “TANNNT” -10 elements and a “TGN” extended -10 elements (see Author response image 6). Considering that only few conserved “TANNNT” motifs in mycobacteria are preceded by an extended -10 element (Cortes et al., 2013; Henry et al., 2020), we speculate that the highly conserved extended -10 element in mycobacterial *carD* promoter may play an important role in maintaining and regulating its basal activity. Moreover, a specific feature (T rich) in the spacer immediately upstream of the extended -10 element contributes greatly to the autoregulation of *Rhodobacter* CarD. In *Mycobacterium*, there is no similar spacer, but there is a highly conserved dinucleotide "CG" immediately upstream of the extended -10 element (see Author response image 6). However, based on this limited information, it is difficult to determine whether mycobacterial CarD negatively regulates its own promoter.

As for the direct transcriptional regulation of the *carD* promoter, the transcriptional factor that can regulate its expression has not yet been identified. By analyzing mycobacterial *carD* promoters, we found two conserved regions upstream of the *carD* core promoter regions (see Author response image 6). We speculate that these sequences may play a role in regulating the expression of *carD*, but to our knowledge, there is no potential transcription factor that can bind to these two sequences. In addition, previous reports showed that the transcription of *carD* is regulated by SigB, but *carD* can still be effectively transcribed in the *sigB* knockout strain (Hurst-Hess et al., 2019). Since the -10 elements recognized by SigA and SigB are somewhat similar in mycobacteria, we speculate that SigA and SigB jointly regulate the *carD* expression, with SigA as the main σ-factor responsible for the basal transcription of *carD*, and SigB as an alternative σ-factor responsible for the stimulated transcription of *carD* under stress conditions, which may also be the reason for the increasing *carD* expression after treatment with DNA-damaging agents (Figure 1A, B).

As for the change of *carD* transcript levels in the absence of AscarD, we have, in fact, shown this result in Figure 5—figure supplement 1, where there is no significant change in *carD* transcription in the AscarD knockdown strain. Furthermore, during this revision process, we successfully constructed an *ascarD* promoter mutant strain that hardly expressed AscarD; when we measured the transcription of *carD* in this mutant strain, we found that the *carD* transcript level did not change significantly. It is currently difficult to explain why the absence of AscarD does not significantly affect the level of *carD* transcription.

**Author response image 6. sa2fig6:** Alignment of mycobacterial *carD* promoter sequences.

6. Can the authors comment on whether the proteolytic degradation of CarD will have an effect on the production of CarD in a feed-back loop?

In our opinion, whether there is a feed-back loop mainly depends on whether CarD is auto-regulated. In *R. sphaeroides*, the expression of *carD* is negatively auto-regulated, so the expressed CarD proteins repress the expression of the *carD* genes and thereby form a feedback loop. Based on existing data, it is difficult to judge whether (or how) mycobacterial CarD is auto-regulated, so it hard to say whether the proteolytic degradation of CarD has an effect on the production of CarD in a feed-back loop.

7. RT-qPCR: No documentation of no RT reactions is given. This is important to exclude dna contamination.

Thank you, we have addressed this problem.

8. Some justification of the use of KatG as a loading control in the western blots is needed, since this is unusual and subject to regulation under stress.

Thank you for this suggestion. We have described this in the Methods and Materials section as follows.

For internal reference protein, we used SigA or KatG as indicated. In the stress stimulation experiments (Figure 1B), SigA was used as an internal control because its level is not affected by the test stimuli. However, SigA protein level in the stationary phase is significantly lower than that in the log phase (Gomez et al., 1998), so when we studied the protein levels in several growth phases, SigA was not used as an internal control. After many tests, we found that the protein level of KatG remained basically unchanged throughout the growth period; therefore, KatG was used as an internal control in experiments at different growth phases. (Note: KatG is highly induced under oxidative stress conditions, so it is not suitable for use as an internal control in the stress stimulation experiments).

9. The rnaseq data presented in the discussion should be moved to results if it is new data. If it is previously published it can be discussed, but the table should be moved to Supplementary information.

Thank you for this advice. The RNA-seq data shown in the discussion was previously published, we have provided a reference and moved it to Supplementary information.

10. A question that arises is whether CarD itself has any effect on the levels of AscarD or whether in published ChIP-seq data, CarD can be seen on the AscarD promoter. Do the authors have any data or comments here?

Thank you for your suggestion. We have analyzed the binding site of CarD through the published ChIP-seq data. As shown in Author response image 7, CarD has strong binding affinity to many promoters, including *lpqE*, *carD* and *cysS*, but has no (or very weak) binding affinity to the *ascarD* promoter. Considering that CarD is recruited to promoters by the RNA polymerase, and that the binding of RNA polymerase to the *ascarD* promoter depends on the function of SigF, we speculate that the small peak of CarD seen at the *ascarD* promoter could be due to the weak transcriptional activity of AscarD under the conditions of the ChIP-seq experiment.

**Author response image 7. sa2fig7:** ChIP-seq reads from *M. smegmatis* DNA coimmunoprecipitated with CarD.

References:

1. Cortes, T., Schubert, O.T., Rose, G., Arnvig, K.B., Comas, I., Aebersold, R., and Young, D.B. (2013). Genome-wide mapping of transcriptional start sites defines an extensive leaderless transcriptome in *Mycobacterium tuberculosis*. Cell Rep *5*, 1121-1131.

2. Henry, K.K., Ross, W., Myers, K.S., Lemmer, K.C., Vera, J.M., Landick, R., Donohue, T.J., and Gourse, R.L. (2020). A majority of *Rhodobacter sphaeroides* promoters lack a crucial RNA polymerase recognition feature, enabling coordinated transcription activation. Proc Natl Acad Sci U S A *117*, 29658-29668.

3. Humpel, A., Gebhard, S., Cook, G.M., and Berney, M. (2010). The SigF regulon in *Mycobacterium smegmatis* reveals roles in adaptation to stationary phase, heat, and oxidative stress. J Bacteriol *192*, 2491-2502.

4. Hurst-Hess, K., Biswas, R., Yang, Y., Rudra, P., Lasek-Nesselquist, E., and Ghosh, P. (2019). Mycobacterial SigA and SigB cotranscribe essential housekeeping genes during exponential growth. mBio *10*, 00273-19.

5. Li, X., Mei, H., Chen, F., Tang, Q., Yu, Z., Cao, X., Andongma, B.T., Chou, S.H., and He, J. (2017). Transcriptome landscape of *Mycobacterium smegmatis*. Front Microbiol *8*, 2505.

6. Oh, Y., Song, S.Y., Kim, H.J., Han, G., Hwang, J., Kang, H.Y., and Oh, J.I. (2020). The partner switching system of the SigF σ factor in *Mycobacterium smegmatis* and induction of the SigF regulon under respiration-inhibitory conditions. Front Microbiol *11*, 588487.

7. Singh, A.K., and Singh, B.N. (2008). Conservation of σ F in mycobacteria and its expression in *Mycobacterium smegmatis*. Curr Microbiol *56*, 574-580.